# Two Pathways to Truthfulness: On the Intrinsic Encoding of LLM Hallucinations

## Abstract

Despite their impressive capabilities, large language models (LLMs) frequently generate hallucinations. Previous work shows that their internal states encode rich signals of truthfulness, yet the origins and mechanisms of these signals remain unclear. In this paper, we demonstrate that truthfulness cues arise from two distinct information pathways: (1) a Question-Anchored pathway that depends on question–answer information flow, and (2) an Answer-Anchored pathway that derives self-contained evidence from the generated answer itself. First, we validate and disentangle these pathways through attention knockout and token patching. Afterwards, we uncover notable and intriguing properties of these two mechanisms and investigate the factors underlying their distinct behaviors. Further experiments reveal that (1) the two mechanisms are closely associated with LLM knowledge boundaries; (2) internal representations are aware of their distinctions; and (3) there is a clear misalignment between truthfulness encoding and language modeling. Finally, building on these insightful findings, two applications are proposed to enhance hallucination detection performance. Overall, our work provides new insight into how LLMs internally encode truthfulness, offering directions for more reliable and self-aware generative systems.

## 1 Introduction

Despite their remarkable capabilities in natural language understanding and generation, large language models (LLMs) often produce *hallucinations*—outputs that appear plausible but are factually incorrect. This phenomenon poses a critical challenge for deploying LLMs in real-world applications where reliability and trustworthiness are paramount (Tian et al., 2024; Shi et al., 2024; Bai et al., 2024). One line of research tackles hallucination detection from an extrinsic perspective (Min et al., 2023; Hu et al., 2025; Huang et al., 2025), evaluating only the model's outputs while disregarding its internal dynamics. Although such approaches can identify surface-level textual inconsistencies, their extrinsic focus limits the insight they offer into the underlying causes of hallucinations. Complementing these efforts, another line of work investigates the intrinsic properties of LLMs, revealing that their internal representations encode rich truthfulness signals (Burns et al., 2023; Li et al., 2023; Chen et al., 2024; Orgad et al., 2025; Niu et al., 2025). These internal truthfulness signals can be exploited to detect an LLM's own generative hallucinations by training a linear classifier (i.e., a probe) on its hidden representations. However, while prior work establishes the presence of such cues, the mechanisms by which they arise and operate remain largely unexplored. Recent studies indicate well-established mechanisms in LLMs that underpin complex capabilities such as in-context learning (Wang et al., 2023), long-context retrieval (Wu et al., 2025), and reasoning (Qian et al., 2025). This observation naturally leads to a key question: *how do truthfulness cues arise and function within LLMs?*

In this paper, we uncover that truthfulness signals in LLMs arise from **two distinct information pathways**: (1) a **Question-Anchored (Q-Anchored) pathway**, which depends on the flow of information from the input question to the generated answer, and (2) an **Answer-Anchored (A-Anchored) pathway**, which derives self-contained evidence directly from the model's own outputs. We begin with a preliminary study using saliency analysis to quantify information flow potentially relevant to hallucination detection. Results reveal a bimodal distribution of dependency on question–answer interactions, suggesting heterogeneous truthfulness encoding mechanisms. To validate this hypothesis, we design two experiments across four diverse datasets and multiple model archi-

tectures and scales, including Llama-3 (1B, 3B, 8B, 70B) (Grattafiori et al., 2024) and Mistral-7B (v0.1, v0.3) (Jiang et al., 2023). By (i) blocking critical question–answer information flow through attention knockout (Geva et al., 2023; Fierro et al., 2025) and (ii) injecting hallucinatory cues into questions via token patching (Ghandeharioun et al., 2024; Todd et al., 2024), we disentangle these distinct truthfulness paths. Our analyses confirm that Q-Anchored signals rely heavily on question-derived cues, whereas A-Anchored signals are robust to their removal and primarily originate from the generated answer itself.

Building on this foundation, we further investigate emergent properties of these truthfulness pathways through large-scale experiments. Our findings highlight several intriguing characteristics: **(1) Association with knowledge boundaries:** Q-Anchored encoding predominates for well-established facts, whereas A-Anchored encoding is favored in uncertain or extrapolated cases. **(2) Self-awareness:** LLM internal states can distinguish which mechanism is being employed, suggesting intrinsic awareness of pathway distinctions and active selection of the appropriate pathway for each instance. **(3) Misalignment with the language modeling objective:** Although truthfulness encoding selectively utilizes question–answer interactions, question–answer attention flows consistently across different encoding mechanisms during generation, underscoring a disconnect between truthfulness encoding and the next-token prediction objective.

Finally, these analyses not only deepen our mechanistic understanding of hallucinations but also enable practical applications. Specifically, two pathway-aware strategies are proposed to enhance hallucination detection: (1) Mixture-of-Probes (MoP): Leveraging the fundamentally different dependencies of the two pathways and the model's intrinsic awareness of pathway selection, MoP employs expert probing classifiers that specialize in different encoding mechanisms. (2) Pathway Reweighting (PR): Inspired by the observed gap between truthfulness encoding and language modeling, PR adjusts information intensity to amplify signals most salient for hallucination detection. Experiments demonstrate that our proposed methods consistently outperform baselines, achieving up to a 10% AUC gain across various datasets and models.

Overall, our key contributions are summarized as follows:

- **(Mechanism)** We conduct a systematic investigation into how internal truthfulness signals emerge and operate within LLMs, revealing two distinct information pathways: a *Question-Anchored* pathway that relies on question–answer information flow, and an *Answer-Anchored* pathway that derives self-contained evidence from the generated output.

- **(Discovery)** Through large-scale experiments across multiple datasets and model families, we identify three key properties of these mechanisms: (i) association with knowledge boundaries, (ii) intrinsic self-awareness enabling pathway selection, and (iii) misalignment between truthfulness encoding and the language modeling objective.

- **(Application)** Building on these findings, we propose two pathway-aware detection methods that exploit the complementary nature of the two mechanisms to enhance hallucination detection, providing new insights for building more reliable generative systems.

## 2 BACKGROUND

### 2.1 HALLUCINATION DETECTION VIA PROBING

We study hallucination detection in question answering. Given an LLM $f$, we denote the dataset as $D = \{(q_i, \hat{y}_i^f, z_i^f)\}_{i=1}^{N}$, where $q_i$ is the question, $\hat{y}_i^f$ the model's generated answer, and $z_i^f \in \{0, 1\}$ indicates whether the answer is hallucinatory. The task is to predict $z_i^f$ given the input $x_i^f = [q_i, \hat{y}_i^f]$ for each instance. Cases in which the model refuses to answer are excluded, as they are not genuine hallucinations and can be trivially classified. Internal–signal methods assume access to the model's hidden representations but no external resources (e.g., retrieval systems or fact–checking APIs). Within this paradigm, probing trains a lightweight linear classifier on hidden activations to distinguish hallucinatory from factual outputs, and has proven highly effective among such methods.

## 2.2 EXACT QUESTION AND ANSWER TOKENS

To analyze the origins and mechanisms of truthfulness signals in LLMs, we focus on **exact tokens** in question–answer pairs. Not all tokens contribute equally to detecting factual errors: some carry core information essential to the meaning of the question or answer, while others provide peripheral details. We draw on *semantic frame* theory (Baker et al., 1998; Pagnoni et al., 2021), which represents a situation or event along with its participants and their roles. In the theory, frame elements are categorized as: (1) *Core frame elements*, which define the situation itself, and (2) *Non-core elements*, which provide additional, non-essential context.

As shown in Table 1, we define: (1) **Exact question tokens:** core frame elements in the question, typically including **the exact subject and property tokens** (i.e., *South Carolina* and *capital*). (2) **Exact answer tokens:** core frame elements in the answer that convey the critical information required to respond correctly (i.e., *Columbia*). Humans tend to rely more on core elements when detecting errors, as these tokens carry the most precise information. Consistent with this intuition, recent work (Orgad et al., 2025) shows that probing activations on the exact answer tokens offers the strongest signal for hallucination detection, outperforming all other token choices. Motivated by these findings, our analysis focuses on exact question and answer tokens and leverages activations of exact answer tokens for probing, enabling a more precise quantification of dependence on question-derived versus answer-internal cues.

---

**Question:** What is the `capital` of `South Carolina` ?

**Answer:** It is `Columbia` , a hub for government, culture, and education that houses the South Carolina State House and the University of South Carolina.

---

Table 1: Example of exact question and answer tokens. Colors indicate token types: ■ – exact property, ■ – exact subject, and ■ – exact answer tokens.

## 3 SELECTIVE DEPENDENCY IN INTERNAL TRUTHFULNESS PATHWAYS

We begin with a preliminary analysis using metrics based on saliency scores (§3.1). The quantitative results reveal **two distinct information pathways for truthfulness encoding:** (1) a **Question-Anchored (Q-Anchored) Pathway**, which relies heavily on exact question tokens, and (2) an **Answer-Anchored (A-Anchored) Pathway**, in which the truthfulness signal is largely independent of the question-to-answer information flow. We further observe that, for different samples, internal representations selectively follow a single distinct pathway, suggesting that LLMs encode truthfulness via multiple heterogeneous mechanisms exclusively. Section 3.2 presents experiments validating this hypothesis. In particular, we show that Q-Anchored Pathway depends critically on information flowing from the question to the answer, whereas the signals along the A-Anchored Pathway are primarily derived from the LLM-generated answer itself.

### 3.1 SALIENCY-DRIVEN PRELIMINARY STUDY

This section investigates the intrinsic characteristics of LLM attention interactions and their potential role in truthfulness encoding. We employ saliency analysis (Simonyan et al., 2014), a widely used interpretability method, to reveal how attention among tokens influences probe decisions. Following common practice (Michel et al., 2019; Wang et al., 2023), we compute the saliency score as:

$$S^l(i,j) = \left| A^l(i,j) \frac{\partial \mathcal{L}(x)}{\partial A^l(i,j)} \right|, \tag{1}$$

where $S^l$ denotes the saliency score matrix of the $l$-th layer, $A^l$ represents the attention weights of that layer, and $\mathcal{L}$ is the loss function for hallucination detection (i.e., the binary cross-entropy loss). Scores are averaged over all attention heads within each layer. In particular, $S^l(i,j)$ quantifies the saliency of attention from query $i$ to key $j$, capturing how strongly the information flow from $j$ to $i$ contributes to the detection. We study two types of information flow: (1) $S_{E_Q \to E_A}$, the saliency of direct information flow from the exact question tokens to the exact answer tokens, and (2) $S_{E_Q \to *}$, the saliency of the total information disseminated by the exact question tokens.

**Results**   We aggregate saliency scores across layers and demonstrate Kernel Density Estimation results on TriviaQA (Joshi et al., 2017) and Natural Questions (Kwiatkowski et al., 2019) datasets. As shown in Figure 1, probability densities reveal a clear bimodal distribution: for all examined information types originating from the question, the probability mass concentrates around two peaks, one near zero saliency and another at a substantially higher value. The near-zero peak suggests that, for a substantial subset of samples, the question-to-answer information flow contributes minimally to hallucination detection, whereas the higher peak reflects strong dependence on such flow.

**Hypothesis**   These observations lead to the hypothesis that there are **two distinct mechanisms of internal truthfulness encoding** for hallucination detection: (1) one characterized by strong reliance on the key question-to-answer information from the exact question tokens, and (2) one in which truthfulness encoding is largely independent of the question. We validate the proposed hypothesis through further experiments in the next section.

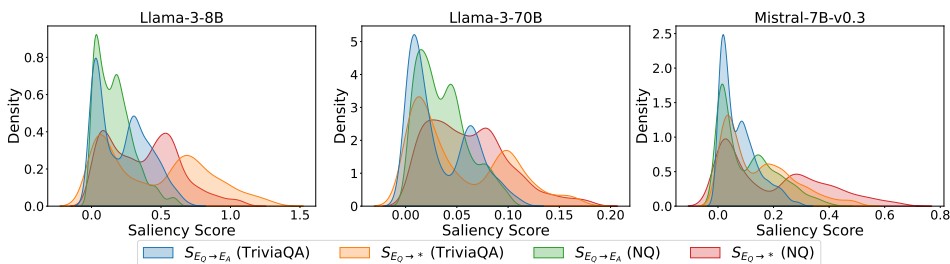

Figure 1: Kernel density estimates of saliency-score distributions for critical question-to-answer information flows. The bimodal pattern reflects two distinct information mechanisms.

### 3.2   DISENTANGLING INFORMATION MECHANISMS

We hypothesize that the internal truthfulness encoding operates through two distinct information flow mechanisms, driven by the attention modules within Transformer blocks. To validate the hypothesis, we first block information flows associated with the exact question tokens and analyze the resulting changes in the probe's predictions. Subsequently, we apply a complementary technique, called token patching, to further substantiate the existence of these two mechanisms. Finally, we demonstrate that the self-contained information from the LLM-generated answer itself drives the truthfulness encoding for the A-Anchored type.

#### 3.2.1   EXPERIMENTAL SETUP

Our analysis spans a diverse set of models varying in scale and architecture, including Llama-3.2-1B, Llama-3.2-3B, Llama-3-8B, Llama-3-70B, Mistral-7B-v0.1, and Mistral-7B-v0.3. We consider four widely used question–answering datasets: PopQA (Mallen et al., 2023), TriviaQA (Joshi et al., 2017), HotpotQA (Yang et al., 2018), and Natural Questions (Kwiatkowski et al., 2019). More details are provided in Appendix B.

#### 3.2.2   IDENTIFYING ANCHORED MODES VIA ATTENTION KNOCKOUT

**Experiment**   To investigate whether internal truthfulness encoding operates via distinct information mechanisms, we perform an attention knockout experiment targeting the exact question tokens. Specifically, for a probe trained on representations from the $k$-th layer, we set $A_l(i, E_Q) = 0$ for all layers $l \in \{1, \ldots, k\}$ and positions $i > E_Q$. This procedure blocks the information flow from question tokens to subsequent positions in the representation. We then examine how the probe's predictions respond to this intervention. To provide a clearer picture, instances are categorized according to whether their prediction $\hat{z}$ changes after the attention knockout:

$$\text{Mode}(x) = \begin{cases} \text{Q-Anchored,} & \text{if } \hat{z} \neq \tilde{\hat{z}} \\ \text{A-Anchored,} & \text{otherwise} \end{cases} \tag{2}$$

where $\hat{z}$ and $\tilde{\hat{z}}$ denote predictions before and after the attention knockout, respectively.

**Results** The results in Figure 2 reveal a clear bifurcation of behaviors: for one subset of instances, probabilities shift substantially, while for another subset, probabilities remain nearly unchanged across all layers. Shaded regions indicate 95% confidence intervals, confirming that this qualitative separation is statistically robust. This sharp divergence supports the hypothesis that internal truthfulness encoding operates via two distinct mechanisms with respect to question–answer information.

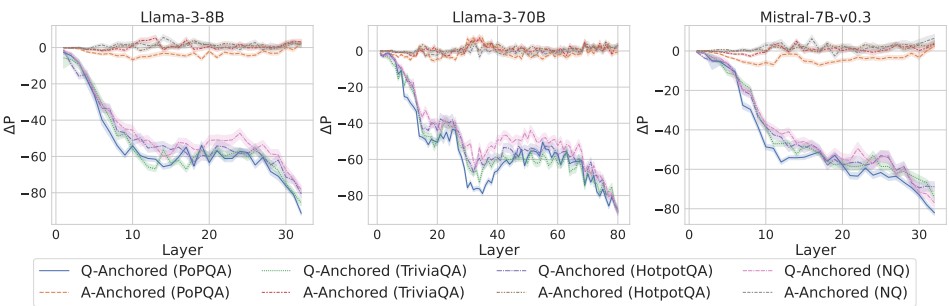

Figure 2: $\Delta$P under attention knockout. The layer axis indicates the Transformer layer on which the probe is trained. Shaded regions indicate 95% confidence intervals. Full results in Figure 7.

### 3.2.3 FURTHER VALIDATION VIA TOKEN PATCHING

**Experiment** To further validate our findings, we employ a critical token patching technique to investigate how the internal representations of the LLM respond to hallucinatory signals originating from exact question tokens under the two proposed mechanisms. Given a context sample $d_c$, we randomly select a patch sample $d_p$ and replace the original question tokens $E_Q^c$ in $d_c$ with the exact question tokens $E_Q^p$ from $d_p$. This operation introduces hallucinatory cues into the context sample, allowing us to assess whether the LLM's internal states appropriately reflect the injected changes. We restrict our analysis to context instances where the original LLM answer is correct and the probe predicts a non-hallucinatory label, ensuring that any observed changes can be attributed solely to the injected hallucinatory cues.

**Results** We measure the sensitivity of the truthfulness signals using the prediction flip rate, defined as the frequency with which the probe's prediction changes after hallucinatory cues are introduced into the exact question tokens. Figure 3 presents the results of the best-performing layer of each model on four datasets when patching the exact subject tokens. Across models and datasets, Q-Anchored mode exhibits significantly higher sensitivity compared to A-Anchored mode when exposed to hallucination cues from the questions. These consistent results provide further support for our hypothesis regarding distinct mechanisms of information pathways.

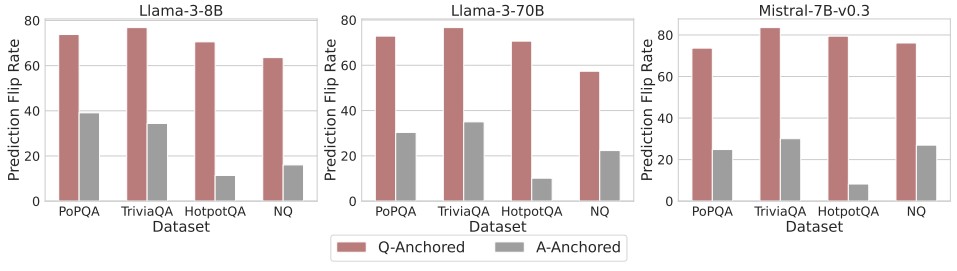

Figure 3: Prediction flip rate under token patching. Q-Anchored samples demonstrate significantly higher sensitivity than the counterparts when hallucinatory cues are injected into exact questions. Full results are in Figures 8 and 9.

### 3.2.4 WHAT DRIVES A-ANCHORED ENCODING?

**Experiment** Since the A-Anchored mode operates largely independently of the question-to-answer information flow, it is important to investigate the source of information it uses to identify

hallucinations. To this end, we remove the questions entirely from each sample, retaining only the LLM-generated answer, and feed this truncated input to the probe. We then evaluate how the probe's predictions change under this "answer-only" condition. This setup enables us to assess whether A-Anchored predictions rely primarily on the generated answer itself rather than on the original question.

**Results**  As shown in Figure 4, Q-Anchored instances exhibit substantial changes in prediction probability when the question is removed, reflecting their dependence on question-to-answer information. In contrast, A-Anchored instances remain largely invariant, indicating that the probe continues to detect hallucinations using information encoded within the LLM-generated answer itself. These findings suggest that the A-Anchored mechanism primarily leverages self-contained answer information to build signals about truthfulness.

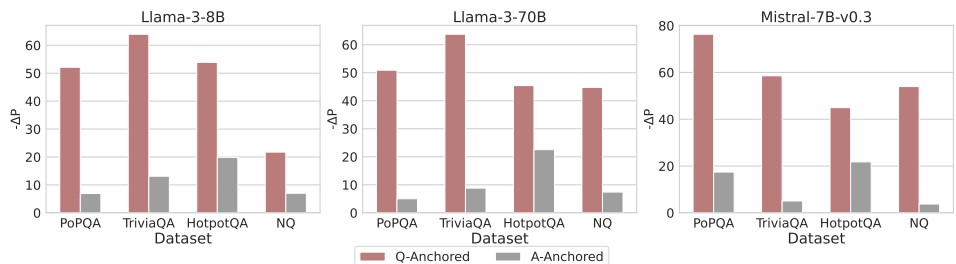

Figure 4: $-\Delta P$ with only the LLM-generated answer. Q-Anchored instances exhibit substantial shifts, whereas A-Anchored instances remain stable, confirming that A-Anchored truthfulness encoding relies on information in the LLM-generated answer itself. Full results in Figures 10, 11.

## 4 EMERGENT PROPERTIES OF INTERNAL TRUTHFULNESS PATHWAYS

This section examines notable properties and distinct behaviors of intrinsic truthfulness encoding: **(1) Associations with knowledge boundaries:** samples within the LLM's knowledge boundary tend to encode truthfulness via the Q-Anchored pathway, whereas samples beyond the boundary often rely on the A-Anchored signal; **(2) Self-awareness:** internal representations can be used to predict which mechanism is being employed, suggesting that LLMs possess intrinsic awareness of pathway distinctions and actively select the appropriate pathway for each instance. **(3) Misalignment between truthfulness encoding and language modeling:** although LLMs exclusively encode truthfulness through different pathways, the intensity of question–answer information flow remains homogeneous during language modeling.

### 4.1 ASSOCIATIONS WITH KNOWLEDGE BOUNDARIES

We find that the distinct behaviors of truthfulness encoding are closely linked to the knowledge boundaries of LLMs. To characterize the boundaries, two complementary metrics are employed: **(1) Answer accuracy**, the most direct indicator of an LLM's factual competence; **(2) Entropy of the exact answer**, a distributional measure capturing the confidence during token generation.

As shown in Figure 5, Q-Anchored samples exhibit significantly higher accuracy than those driven by the A-Anchored pathway and generally display lower entropy, indicating greater confidence in the predicted answer. These findings suggest that truthfulness encoding aligns strongly with the presence or absence of stored knowledge: when LLMs possess the necessary knowledge, they tend to rely on question–answer information flow (Q-Anchored); when uncertain, they instead draw on internal patterns within their own generated output (A-Anchored).

### 4.2 INTRINSIC AWARENESS OF ENCODING PATHWAY SELECTION

Given that LLMs encode truthfulness signals through two distinct mechanisms, this section explores whether their internal representations contain information indicative of the selective cues. To this

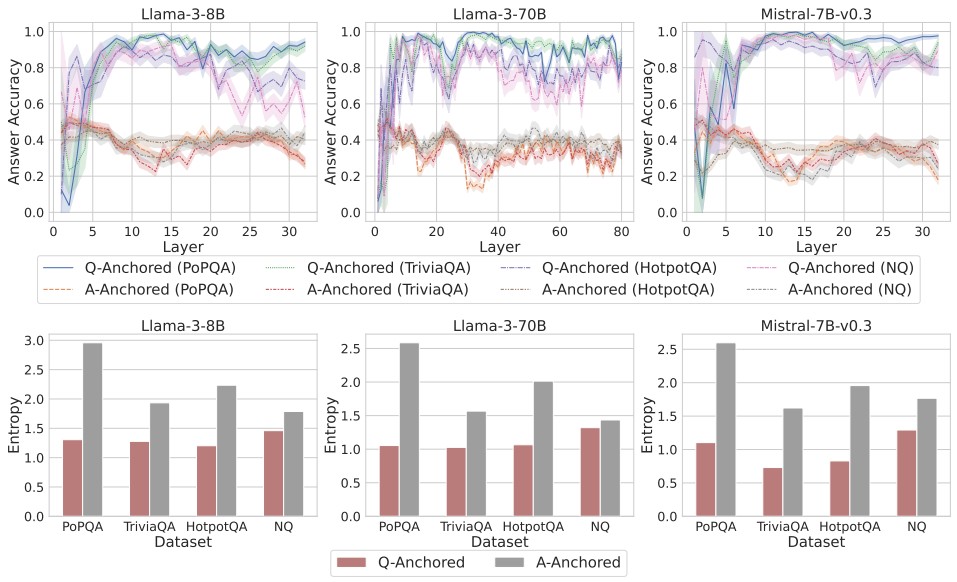

Figure 5: Comparisons of accuracy and entropy between pathways. Q-Anchored samples show higher accuracy and lower entropy than A-Anchored ones, highlighting the link between truthfulness encoding and LLM knowledge boundaries. Full results in Figures 12 and 13.

end, we train probing classifiers from the model's internal states to predict which mechanism is being employed.

Table 2 reports the pathway classification results of the best-performing layers in hallucination detection across different models. Our findings demonstrate that the pathway selection can be reliably inferred from internal representations, suggesting that, in addition to encoding answer truthfulness, LLMs exhibit intrinsic awareness of pathway distinctions and actively select between them. These findings highlight a potential avenue for fine-grained improvements targeting specific truthfulness encoding mechanisms.

Table 2: AUCs for encoding pathway classification. The predictability of pathway selection from internal representations indicates that LLMs possess intrinsic awareness of pathway distinctions.

| Datasets | Llama-3-8B | Llama-3-70B | Mistral-7B-v0.1 | Mistral-7B-v0.3 |
|---|---|---|---|---|
| PopQA | 87.80 | 92.66 | 85.16 | 87.64 |
| TriviaQA | 75.10 | 83.91 | 79.62 | 85.87 |
| HotpotQA | 86.31 | 87.34 | 91.19 | 92.13 |
| NQ | 78.31 | 84.14 | 81.14 | 84.83 |

## 4.3 MISALIGNMENT WITH LANGUAGE MODELING

Although LLMs encode truthfulness through two distinct internal pathways, our analysis reveals a clear disconnect between this truthfulness encoding and the surface-level language modeling process. We assess this relationship by computing the attention weights assigned to the exact question tokens during generation of the exact answer tokens, which captures the intensity of question–answer information flow that drives prediction of the exact answer.

As shown in Figure 6, the overall distribution of question–answer attention remain largely uniform across both Q-Anchored and A-Anchored cases. This consistency indicates that LLMs sustain a stable question–answer information flow regardless of which truthfulness pathway is engaged. Such invariance underscores a fundamental misalignment: while the hidden representations contain rich, pathway-specific truthfulness cues, the next-token prediction dynamics remain agnostic to these signals. This gap may potentially stem from the training objective of LLMs, which prioritizes gener-

ating plausible continuations over explicitly modeling factual correctness (Zhang et al., 2025; Kalai et al., 2025).

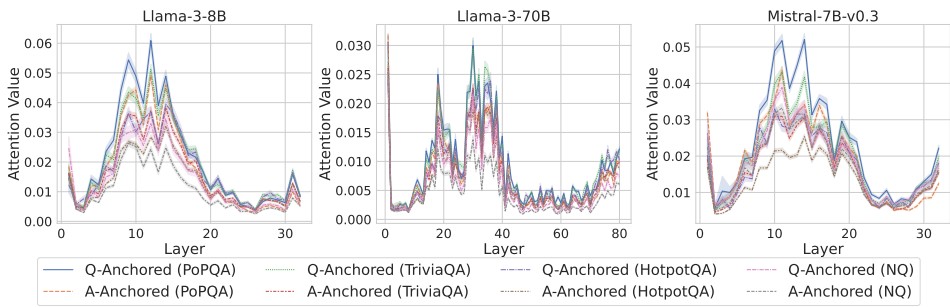

Figure 6: Question–answer information intensity for Q-Anchored and A-Anchored samples. The distribution remain homogeneous across truthfulness pathways, revealing that the surface-level language modeling is insensitive to pathway-specific truthfulness encoding. Full results in Figure 14.

## 5 APPLICATIONS: PATHWAY-AWARE HALLUCINATION DETECTION

Building on the intriguing findings, we explore how the discovered pathway distinctions can be leveraged to improve hallucination detection. Specifically, two simple yet effective pathway-aware strategies are proposed: **(1) Mixture-of-Probes (MoP)** (§5.1), which allows expert probes to specialize in Q-Anchored and A-Anchored pathways respectively, and **(2) Pathway Reweighting (PR)** (§5.2), a plug-and-play approach that amplifies pathway-relevant cues salient for detection.

### 5.1 MIXTURE-OF-PROBES

Motivated by the fundamentally different dependencies of the two encoding pathways and the LLMs' intrinsic awareness of pathway selection, we propose a **Mixture-of-Probes (MoP)** framework that explicitly captures this heterogeneity. Rather than training a single probe to handle all inputs, MoP employs two pathway-specialized experts and leverages the self-awareness probe (§4.2) as a gating network to combine their predictions. Let $\mathbf{h}^{l^*}(x) \in \mathbb{R}^d$ be the hidden state of the exact answer token from the best detection layer $l^*$. Two expert probes $p_Q(\cdot)$ and $p_A(\cdot)$ are trained separately for two pathway samples, and the self-awareness probe (§4.2) provides a gating coefficient $\pi(\mathbf{h}^{l^*}(x)) \in [0, 1]$. The final prediction is a convex combination, requiring no extra training:

$$p_{\text{MoP}}(z=1 \mid \mathbf{h}^{l^*}(x)) = \pi_Q \, p_Q(z=1 \mid \mathbf{h}^{l^*}(x)) + (1 - \pi_Q) \, p_A(z=1 \mid \mathbf{h}^{l^*}(x)). \tag{3}$$

### 5.2 PATHWAY REWEIGHTING

Inspired by the observed gap between the intrinsic truthfulness cues and the next-token prediction dynamics, we introduce a plug-and-play **Pathway Reweighting (PR)** method that directly modulates the question–answer information flow. The key idea is to adjust the attention from exact answer to question tokens according to the predicted pathway, thereby amplifying the signals most salient for hallucination detection. For each layer $l \leq l^*$, two learnable scalars $\alpha_Q^l, \alpha_A^l > 0$ are introduced. Given pathway selection probability $\pi(\mathbf{h}^{l^*}(x))$, we rescale attention edges $i \in E_A, j \in E_Q$:

$$\tilde{A}^l(i, j) = \begin{cases} \left[1 + s(\mathbf{h}^{l^*}(x))\right] A^l(i, j), & i \in E_A, j \in E_Q, \\ A^l(i, j), & \text{otherwise,} \end{cases} \tag{4}$$

where

$$s(\mathbf{h}^{l^*}(x)) = \pi_Q \, \alpha_Q^l - (1 - \pi_Q) \, \alpha_A^l. \tag{5}$$

The extra parameters serve as a lightweight adapter, used only during detection to guide salient truthfulness cues and omitted during generation, leaving the generation capacity unaffected.

## 5.3 EXPERIMENTS

**Setup** The experimental setup follows Section 3.2.1. We compare our method against several internal-based baselines, including (1) P(True) (Kadavath et al., 2022), (2) uncertainty-based metrics (Aichberger et al., 2024; Xue et al., 2025a), and (3) probing classifiers (Chen et al., 2024; Orgad et al., 2025). Results are averaged over three random seeds. Additional implementation details are provided in Appendix B.4 and B.5.

**Results** As shown in Table 3, both MoP and PR consistently outperform standard baselines across datasets and model scales. MoP leverages pathway-specialized experts, while PR further improves performance by dynamically shifting the focus on the selected truthfulness cues. These results demonstrate that explicitly modeling truthfulness encoding heterogeneity can effectively translate the insights of our analysis into practical gains for hallucination detection.

Table 3: Comparison of hallucination detection performance (AUC). Full results in Tables 7, 8, 9.

| Method | Llama-3-8B | | | | Mistral-7B-v0.3 | | | |
|---|---|---|---|---|---|---|---|---|
| | PopQA | TriviaQA | HotpotQA | NQ | PopQA | TriviaQA | HotpotQA | NQ |
| P(True) | 55.85 | 49.92 | 52.14 | 53.27 | 45.49 | 47.61 | 57.87 | 52.79 |
| Logits-mean | 74.52 | 60.39 | 51.94 | 52.63 | 69.52 | 66.76 | 55.45 | 57.88 |
| Logits-min | 85.36 | 70.89 | 61.28 | 56.50 | 87.05 | 77.33 | 68.08 | 54.40 |
| Probing Baseline | 88.71 | 77.58 | 82.23 | 70.20 | 87.39 | 81.74 | 83.19 | 73.60 |
| MoP | 92.11 | 81.18 | 85.45 | 74.64 | 91.66 | 83.57 | 85.82 | 76.87 |
| PR | **94.01** | **83.13** | **87.81** | **79.10** | **93.09** | **84.36** | **89.03** | **79.09** |

## 6 RELATED WORK

Hallucination detection in LLMs has received increasing attention because of its critical role in building reliable and trustworthy generative systems (Tian et al., 2024; Shi et al., 2024; Bai et al., 2024). Existing approaches can be broadly grouped by whether they rely on external resources (e.g., retrieval systems or fact–checking APIs). Externally assisted methods cross-verify output texts against external knowledge bases (Min et al., 2023; Hu et al., 2025; Huang et al., 2025) or specialized LLM judges (Luo et al., 2024; Bouchard & Chauhan, 2025; Zhang et al., 2025). Resource-free methods avoid external data and instead exploit the model's own intermediate computations. Some leverage the model's self-awareness of knowledge boundaries (Kadavath et al., 2022), while others use uncertainty-based measures (Aichberger et al., 2024; Xue et al., 2025a), treating confidence as a proxy for truthfulness. These techniques analyze output distributions (e.g., logits) (Aichberger et al., 2024), variance across multiple samples (e.g., consistency) (Min et al., 2023; Aichberger et al., 2025), or other statistical indicators of prediction uncertainty (Xue et al., 2025b). Another line of work trains linear probing classifiers on hidden representations to capture intrinsic truthfulness signals. Prior work (Burns et al., 2023; Li et al., 2023; Chen et al., 2024; Orgad et al., 2025) shows that LLMs encode rich latent features correlated with factual accuracy, enabling efficient detection with minimal overhead. Yet the mechanisms behind these internal truthfulness encoding remain poorly understood. Compared to previous approaches, our work addresses this gap by dissecting how such intrinsic signals emerge and operate, revealing distinct information pathways that not only yield explanatory insights but also enhance detection performance.

## 7 CONCLUSION

We investigate how LLMs encode truthfulness, revealing two complementary pathways: a *Question-Anchored* pathway relying on question–answer flow, and an *Answer-Anchored* pathway extracting self-contained evidence from generated outputs. Analyses across datasets and models highlight their ties to knowledge boundaries, intrinsic self-awareness, and partial misalignment with the language modeling objective. Building on these insights, we further propose two applications to improve hallucination detection. Overall, our findings advance understanding of intrinsic truthfulness encoding and offer practical applications for building more reliable and self-aware generative systems.

ETHICS STATEMENT

Our work presents minimal potential for negative societal impact, primarily due to the use of publicly available datasets and models. This accessibility inherently reduces the risk of adverse effects on individuals or society.

REPRODUCIBILITY STATEMENT

We are committed to ensuring the reproducibility of our results. To facilitate replication, we provide a comprehensive description of our methods and experimental setup in the main text and Appendix B. Our paper specifies model architectures, datasets, data splits, and all preprocessing steps in detail. Hyperparameters, training procedures, and evaluation metrics are also reported in both the main text and Appendix B. All experiments rely solely on publicly available models and datasets, requiring no proprietary resources. This documentation supplies all information necessary for researchers to reproduce our findings.

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

## A    LLM USAGE

In this work, we employ LLMs solely for language refinement to enhance clarity and explanatory quality. All content has been carefully verified for factual accuracy, and the authors take full responsibility for the entire manuscript. The core ideas, experimental design, and methodological framework are conceived and developed independently by the authors, without the use of LLMs.

## B    IMPLEMENTATION DETAILS

### B.1    IDENTIFYING EXACT QUESTION AND ANSWER TOKENS

To locate the exact question and answer tokens within a QA pair, we prompt GPT-4o (version `gpt-4o_2024-11-20`) to identify the precise positions of the core frame elements. The instruction templates are presented in Tables 5 and 6. A token is considered an *exact question* or *exact answer* if and only if it constitutes a valid substring of the corresponding question or answer. To mitigate potential biases, each example is prompted at most five times, and only successfully extracted instances are retained for downstream analysis.

### B.2    PROBING IMPLEMENTATION DETAILS

We perform probing on the intermediate representations of the *last exact answer token*, as previous work (Orgad et al., 2025) has shown that this token encodes the strongest truthfulness signals. Specifically, we use the activations from the final multi-layer perceptron (MLP) of each layer. Our experiments indicate that using MLP outputs or attention outputs yields similar results. For the probing classifier, we follow standard practice (Chen et al., 2024; Orgad et al., 2025) and employ a logistic regression model implemented in `scikit-learn`.

### B.3    DATASETS

We consider four widely used question–answering datasets: PopQA (Mallen et al., 2023), TriviaQA (Joshi et al., 2017), HotpotQA (Yang et al., 2018), and Natural Questions (Kwiatkowski et al., 2019).

**PopQA** is an open-domain question-answering dataset that emphasizes entity-centric factual knowledge with a long-tail distribution. It is designed to probe LLMs' ability to memorize less frequent facts, highlighting limitations in parametric knowledge. **TriviaQA** is a reading comprehension dataset constructed by pairing trivia questions authored independently of evidence documents. The questions are often complex, requiring multi-sentence reasoning, and exhibit substantial lexical and syntactic variability. **HotpotQA** is a challenging multi-hop question-answering dataset that requires reasoning. It includes diverse question types—span extraction, yes/no, and novel comparison questions—along with sentence-level supporting fact annotations, promoting the development of explainable QA systems. **Natural Questions** is an open-domain dataset consisting of real, anonymized questions from Google search queries. Each question is annotated with both a long answer (paragraph or section) and a short answer (span or yes/no), or marked as null when no answer is available. Due to computational constraints, we randomly sample 2,000 training samples and 2,000 test samples for each dataset.

### B.4    APPLICATIONS: IMPLEMENTATION DETAILS OF BASELINES

In our experiments regarding applications, we compare our proposed methods against several internal-based baselines for hallucination detection. These baselines leverage the LLM's internal signals, such as output probabilities, logits, and hidden representations, without relying on external resources. Below, we detail the implementation of each baseline.

**P(True)**    P(True) (Kadavath et al., 2022) exploits the LLM's self-awareness of its knowledge boundaries by prompting the model to assess the correctness of its own generated answer. Specifically, for each question-answer pair $(q_i, \hat{y}_i^f)$, we prompt the LLM with a template that asks it to evaluate whether its answer is factually correct. Following Kadavath et al. (2022), the prompt template is shown in Table 4.

Table 4: Prompt template used for the P(True) baseline.

| |
|---|
| Question: {Here is the question} |
| Possible answer: {Here is the answer} |
| Is the possible answer: |
| (A) True |
| (B) False |
| The possible answer is: |

**Logits-based Baselines** The logits-based baselines utilize the raw logits produced by the LLM during the generation of the exact answer tokens. Let $\hat{y}_{i,E_A}^f = [t_1, t_2, \ldots, t_m]$ represent the sequence of exact answer tokens for a given question-answer pair, where $m$ is the number of exact answer tokens. For each token $t_j$ (where $j \in \{1, \ldots, m\}$), the LLM produces a logit vector $L_j \in \mathbb{R}^V$, where $V$ is the vocabulary size, and the logit for the generated token $t_j$ is denoted $L_j[t_j]$. The logits-based metrics are defined as follows:

- **Logits-mean**: The average of the logits across all exact answer tokens:

$$\text{Logits-mean} = \frac{1}{m} \sum_{j=1}^{m} L_j[t_j]$$

- **Logits-max**: The maximum logit value among the exact answer tokens:

$$\text{Logits-max} = \max_{j \in \{1,\ldots,m\}} L_j[t_j]$$

- **Logits-min**: The minimum logit value among the exact answer tokens:

$$\text{Logits-min} = \min_{j \in \{1,\ldots,m\}} L_j[t_j]$$

These metrics serve as proxies for the model's confidence in the generated answer, with lower logit values potentially indicating uncertainty or hallucination.

**Scores-based Baselines** The scores-based baselines are derived from the softmax probabilities of the exact answer tokens. Using the same notation as above, for each exact answer token $t_j$, the softmax probability is computed as:

$$p_j[t_j] = \frac{\exp(L_j[t_j])}{\sum_{k=1}^{V} \exp(L_j[k])}$$

where $L_j[k]$ is the logit for the $k$-th token in the vocabulary. The scores-based metrics are defined as follows:

- **Scores-mean**: The average of the softmax probabilities across all exact answer tokens:

$$\text{Scores-mean} = \frac{1}{m} \sum_{j=1}^{m} p_j[t_j]$$

- **Scores-max**: The maximum softmax probability among the exact answer tokens:

$$\text{Scores-max} = \max_{j \in \{1,\ldots,m\}} p_j[t_j]$$

- **Scores-min**: The minimum softmax probability among the exact answer tokens:

$$\text{Scores-min} = \min_{j \in \{1,\ldots,m\}} p_j[t_j]$$

These probabilities provide a normalized measure of the model's confidence, bounded between 0 and 1, with lower values potentially indicating a higher likelihood of hallucination.

**Probing Baseline**   The probing baseline follows the standard approach described in Chen et al. (2024); Orgad et al. (2025). A linear classifier is trained on the hidden representations of the last exact answer token from the best-performing layer. The classifier is implemented using `scikit-learn` with default hyperparameters, consistent with the probing setup described in Section B.2. The probing baseline serves as a direct comparison to our proposed applications, as it relies on the same type of internal signals but does not account for the heterogeneity of truthfulness encoding pathways.

### B.5 APPLICATIONS: IMPLEMENTATION DETAILS OF OUR METHODS

**Model Backbone and Hidden Representations**   All experiments use the same base LLM as in the main paper. Hidden representations $\mathbf{h}^{l^*}(x)$ are extracted from the best-performing layer $l^*$ determined on a held-out validation split.

**Mixture-of-Probes (MoP)**   The two expert probes $p_Q$ and $p_A$ are implemented as vanilla logistic regression classifiers using `scikit-learn` with default hyperparameters. The gating network is from the self-awareness probe described in Section 4.2. The proposed MoP framework requires no additional retraining: we directly combine the two expert probes with the pathway-selection classifier described in Section 4.2 and perform inference without further parameter updates.

**Pathway Reweighting (PR)**   For each Transformer layer $l \leq l^*$, we introduce learnable scalars $\alpha_Q^l$ and $\alpha_A^l$ for every attention head. These parameters are optimized using the Adam optimizer with a learning rate of $1\mathrm{e}{-2}$, a batch size of 512, and a total of 10 training epochs, while keeping all original LLM parameters frozen.

Table 5: Prompt template used to locate the exact question tokens.

You are given a factual open-domain question-answer pair.
Your task is to identify:

1. Core Entity (c) - the known specific entity in the question that the answer is about (a person, place, organization, or other proper noun).
2. Relation (r) - the minimal phrase in the question that expresses what is being asked about the core entity, using only words from the question.

Guidelines:

The core entity must be a concrete, known entity mentioned in the question, not a general category.
If multiple entities appear, choose the one most central to the question—the entity the answer primarily concerns.
The relation should be the smallest meaningful span that directly connects the core entity to the answer.
Use only words from the question; do not paraphrase or add new words.
Exclude extra context, modifiers, or descriptive phrases that are not essential to defining the relationship.
For complex questions with long modifiers or embedded clauses, focus on the words that directly express the property, action, or attribute of the core entity relevant to the answer.
If you cannot confidently identify the core entity or the relation, output NO ANSWER.

Output format:
Core Entity: exact text
Relation: exact text

Example 1
Question: Who was the director of Finale?
Answer: Ken Kwapis
Core Entity: Finale
Relation: director

Example 2
Question: What film, in production between 2007 and 2009, is directed by James Cameron ("Titanic")?
Answer: Avatāra
Core Entity: James Cameron
Relation: film directed by

Example 3
Question: Which novel, written in 1925 and often cited as a classic of American literature, was authored by F. Scott Fitzgerald?
Answer: The Great Gatsby
Core Entity: F. Scott Fitzgerald
Relation: novel authored by

Question: {Here is the question}
Answer: {Hese is the answer}

Table 6: Prompt template used to locate the exact answer tokens.

You are given a factual open-domain question-answer pair.
Your task is to extract the short exact answer from the long answer.

Guidelines:

Retain only the relevant tokens from the answer.
Use only words from the answer; do not paraphrase or add new words.
Extract the short answer regardless of whether the long answer is correct or incorrect.
If the short answer cannot be confidently determined, or if the long answer does not address the question, output NO ANSWER.

Example 1
Q: Which musical featured the song The Street Where You Live?
A: The song "The Street Where You Live" is from the Lerner and Loewe musical "My Fair Lady." It is one of the most famous songs from the show, and it is sung by Professor Henry Higgins as he reflects on the transformation of Eliza Doolittle and the memories they have shared together.
Exact answer: My Fair Lady

Example 2
Q: Which Swedish actress won the Best Supporting Actress Oscar for Murder on the Orient Express?
A: I'm glad you asked about a Swedish actress who won an Oscar for "Murder on the Orient Express," but I must clarify that there seems to be a misunderstanding here. No Swedish actress has won an Oscar for Best Supporting Actress for that film. The 1974 "Murder on the Orient Express" was an American production, and the cast was predominantly British and American. If you have any other questions or if there's another
Exact answer: NO ANSWER

Example 3
Q: Who invented the telephone?
A: The telephone was invented by Thomas Edison in the 19th century.
Exact answer: Thomas Edison

Q: {Here is the question}
A: {Here is the answer}
Exact answer:

## C FULL RESULTS

### C.1 FULL ATTENTION KNOCKOUT RESULTS

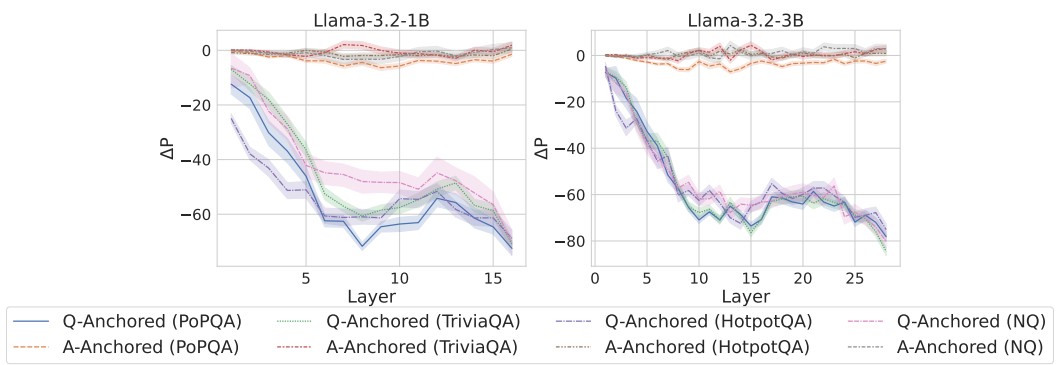

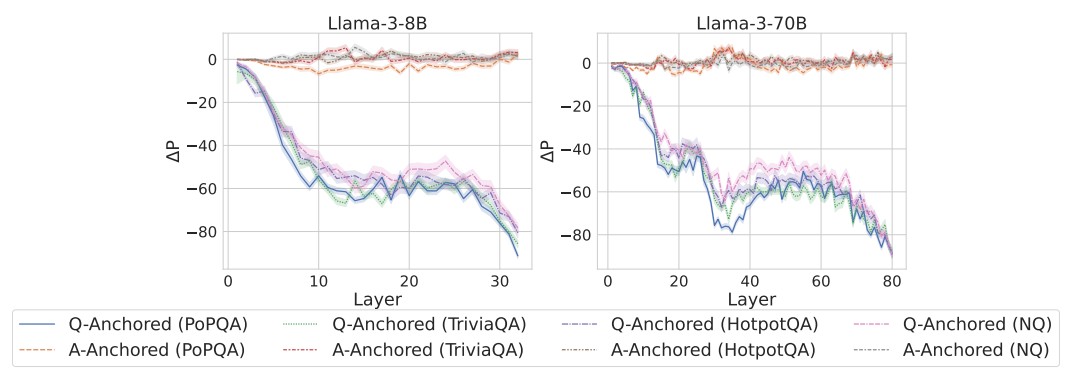

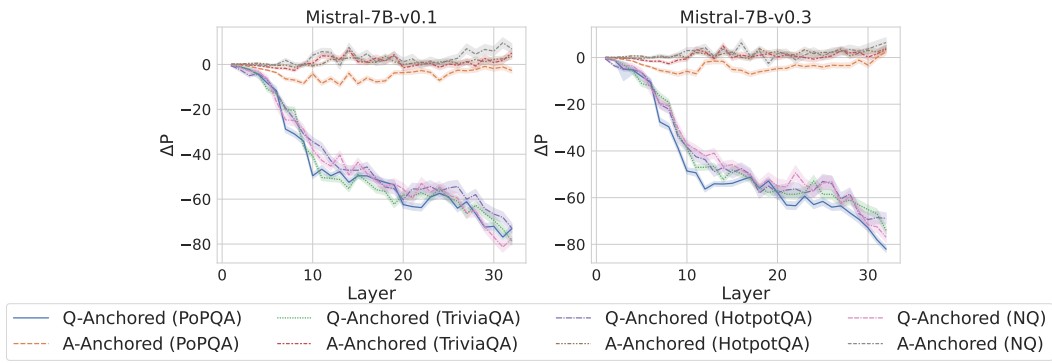

Figure 7: Full results under attention knockout.

## C.2 Full Token Patching Results

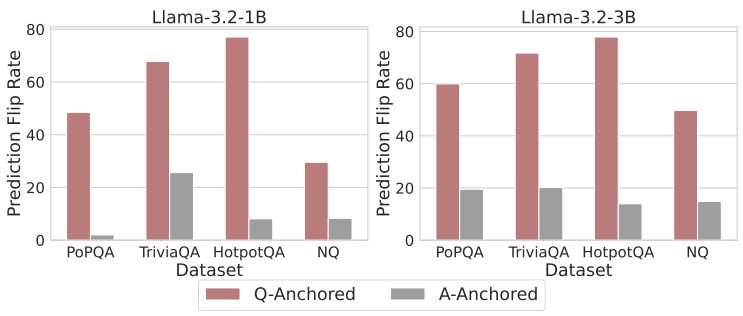

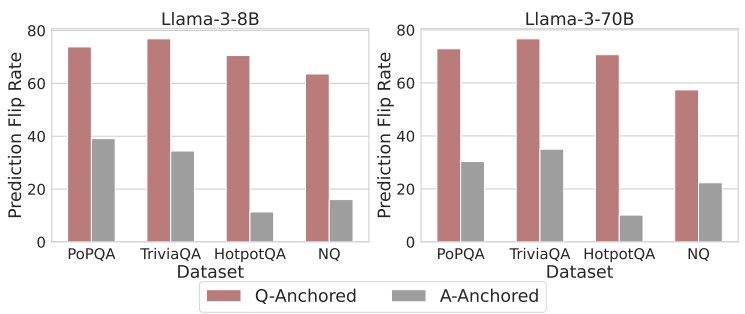

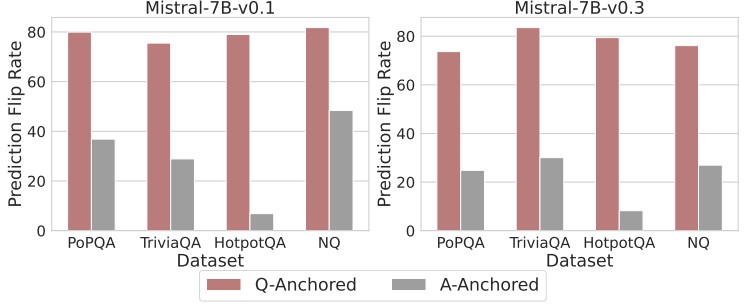

Figure 8: Full results under token patching for the best-performing layers.

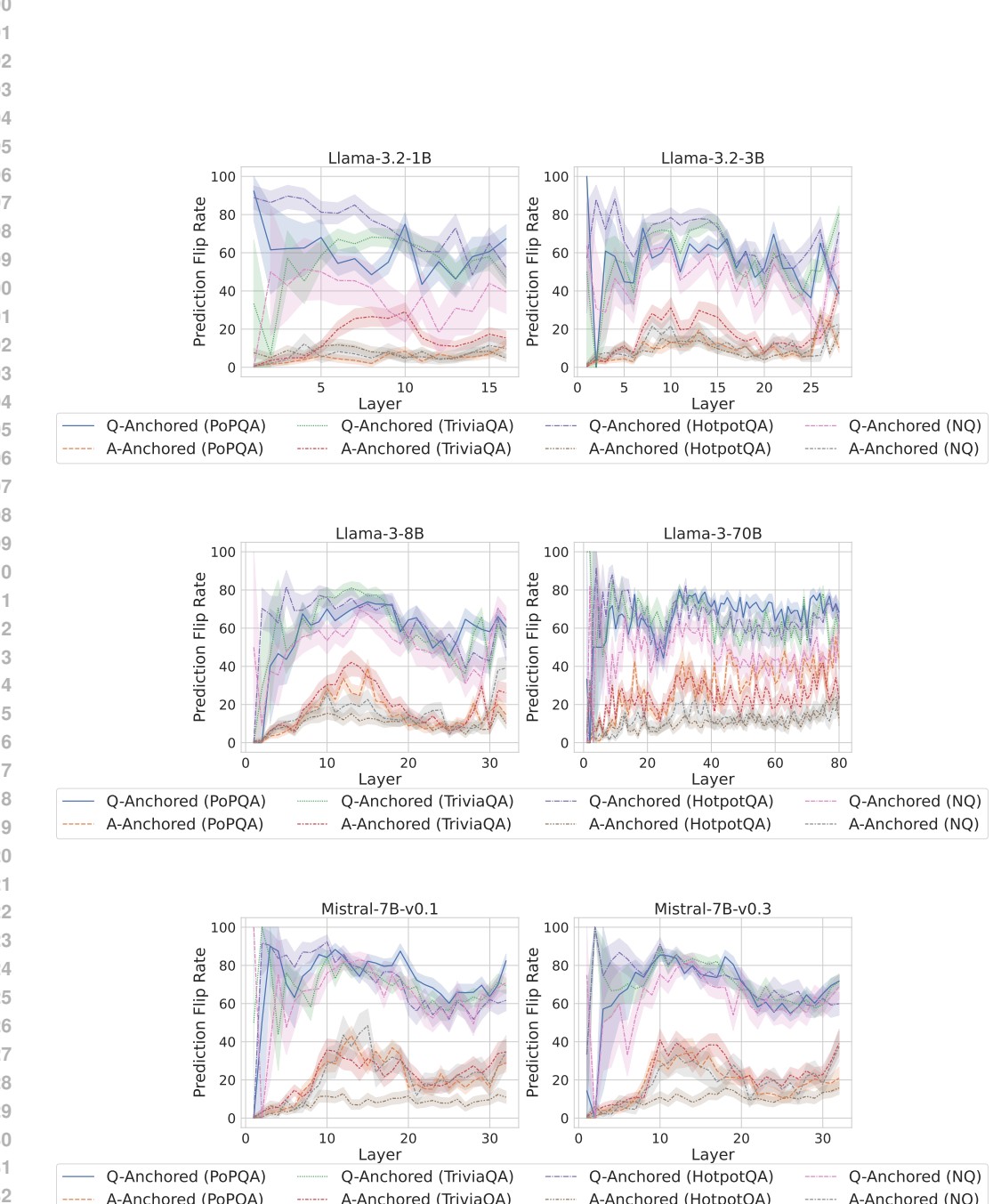

Figure 9: Full results under token patching for the all layers.

## C.3 FULL ANSWER-ONLY RESULTS

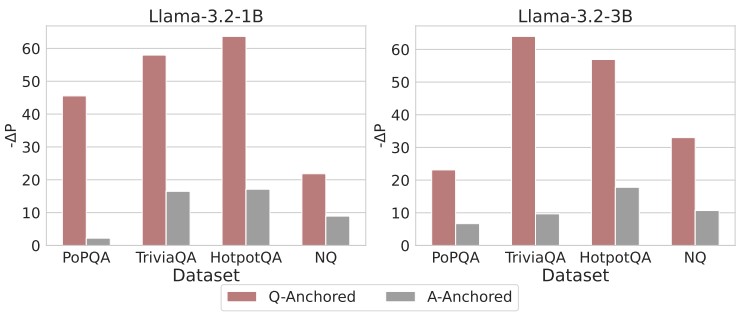

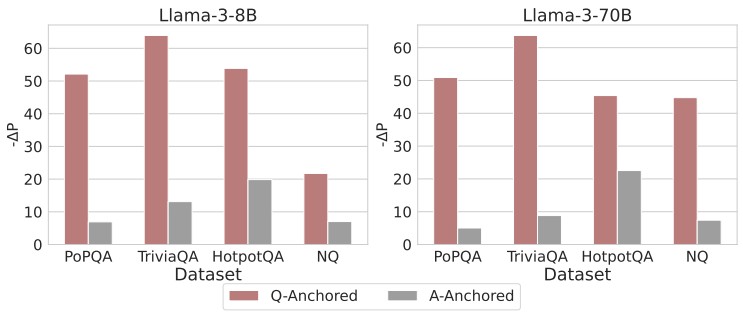

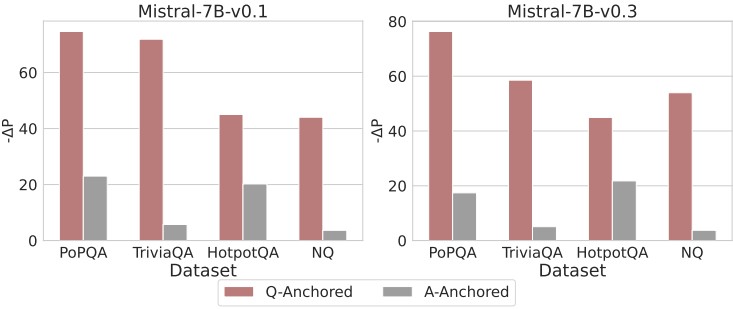

Figure 10: Full results for the best-performing layers when only the LLM-generated answer is provided.

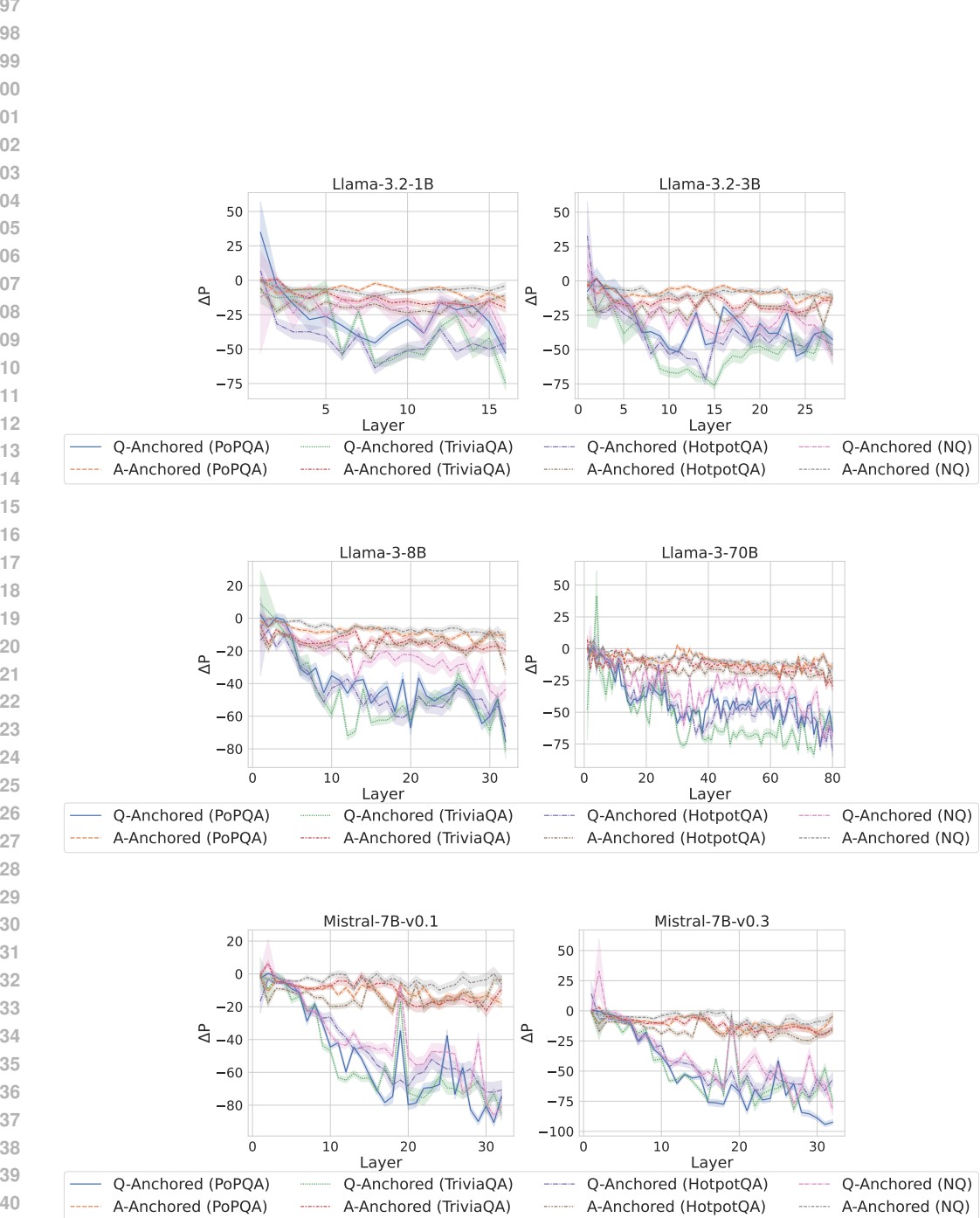

Figure 11: Full results for all layers when only the LLM-generated answer is provided.

## C.4 FULL RESULTS REGARDING KNOWLEDGE BOUNDARIES

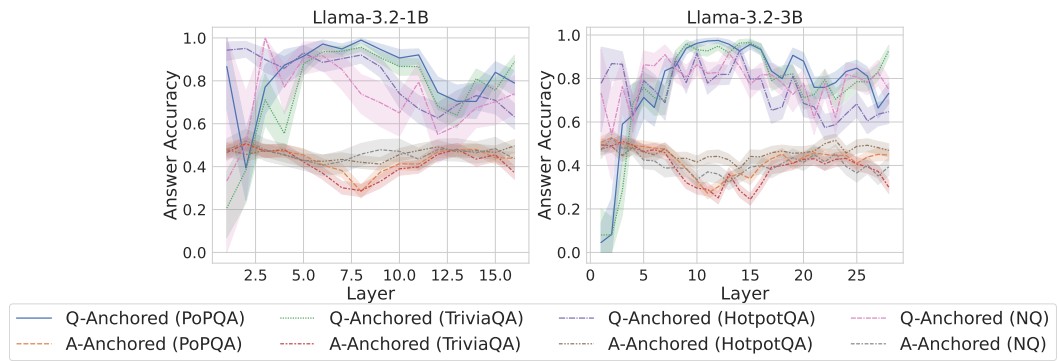

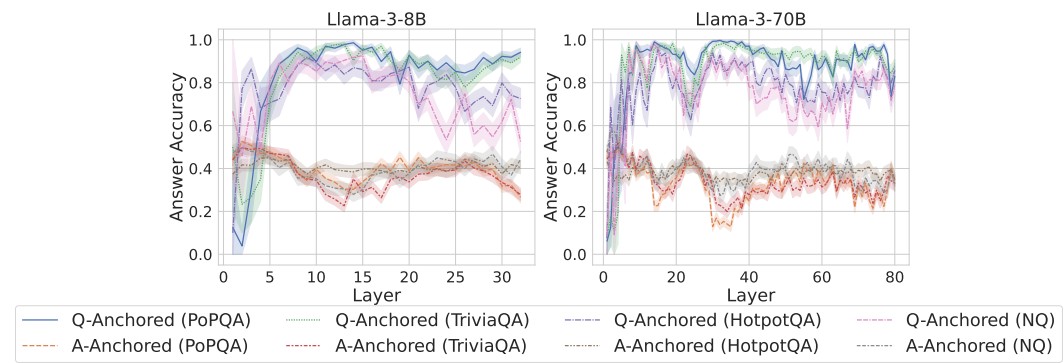

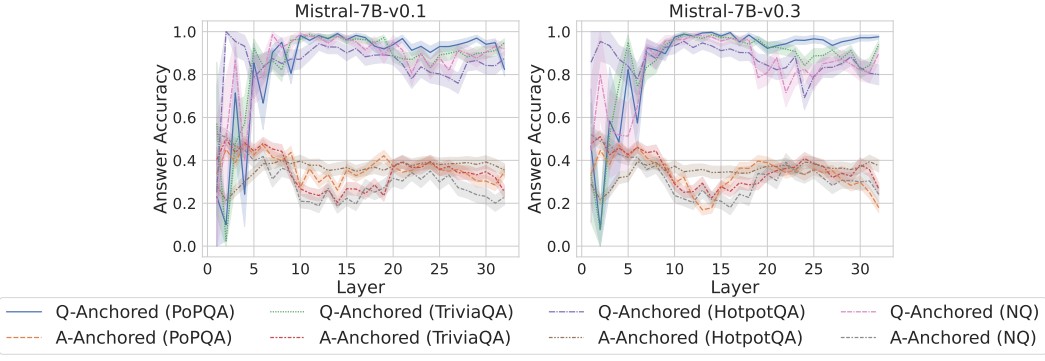

Figure 12: Full results for comparisons of answer accuracy between pathways.

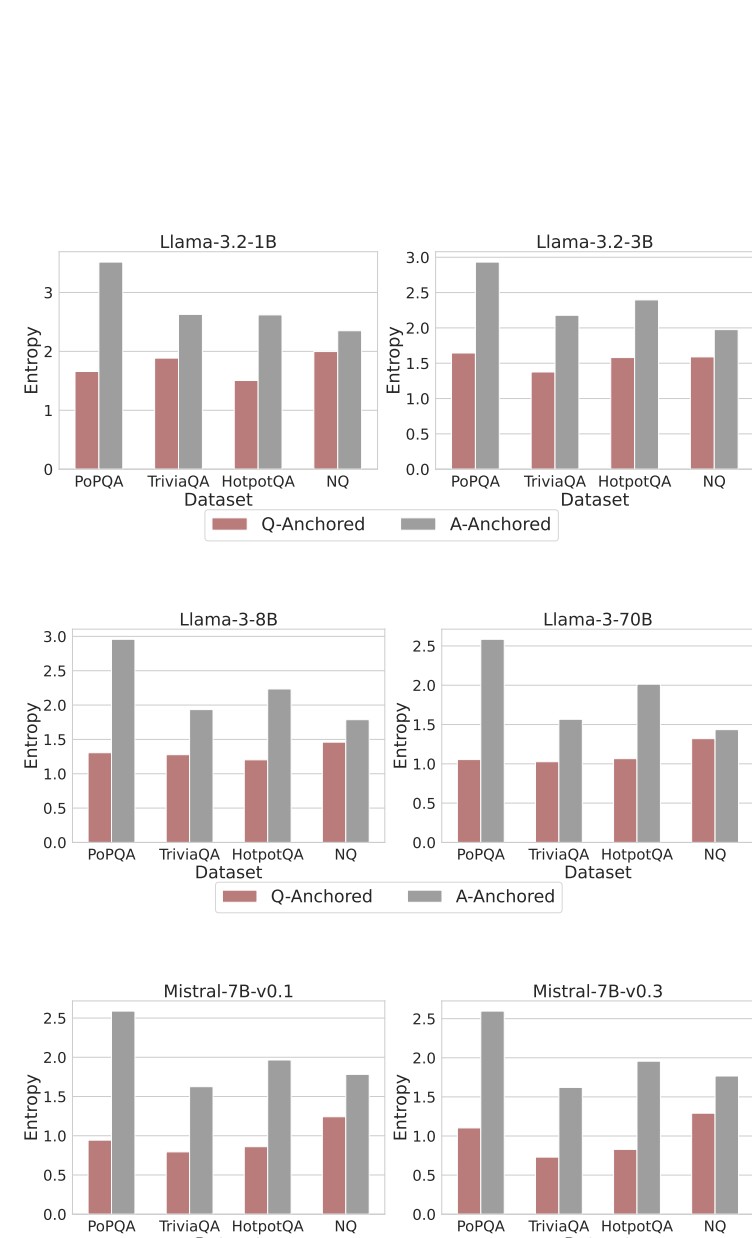

Figure 13: Full results for comparisons of entropy between pathways.

## C.5 FULL RESULTS REGARDING QUESTION–ANSWER INFORMATION INTENSITY

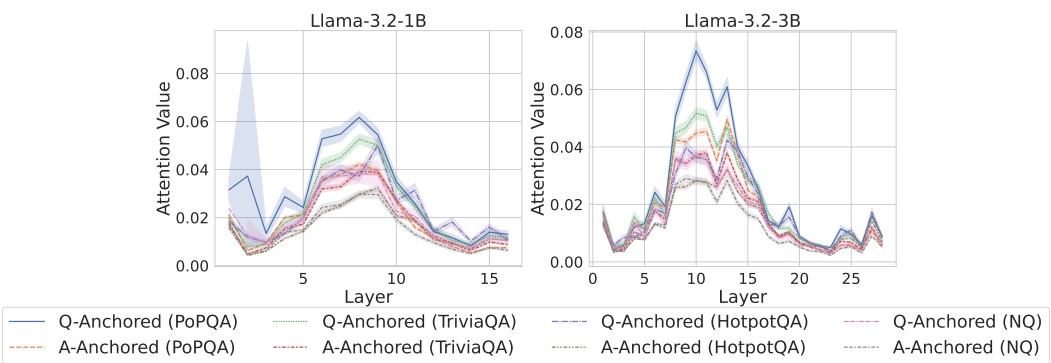

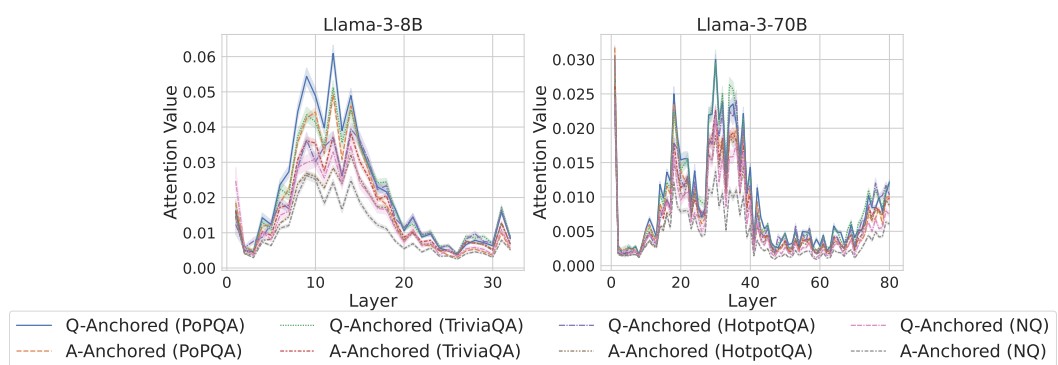

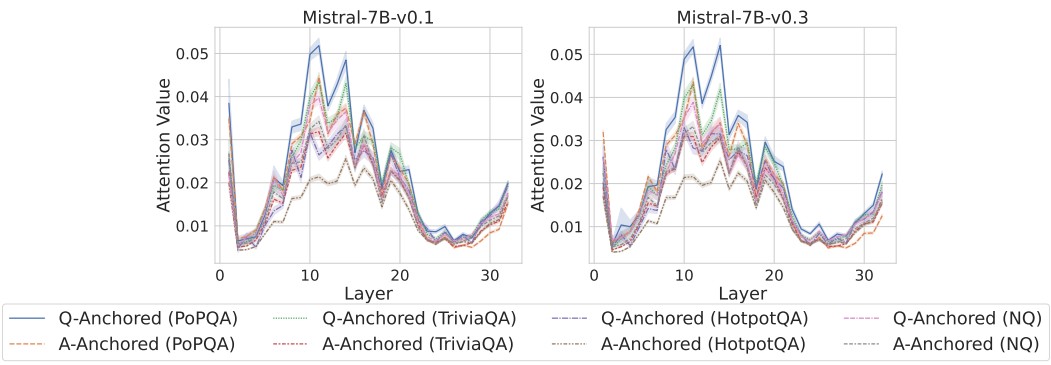

Figure 14: Full results for question–answer information intensity.

## C.6 FULL RESULTS REGARDING APPLICATIONS

Table 7: Comparison of hallucination detection performance (AUC) on LLama-3.2-1B and LLama-3.2-3B.

| Method | LLama-3.2-1B | | | | LLama-3.2-3B | | | |
|---|---|---|---|---|---|---|---|---|
| | PopQA | TriviaQA | HotpotQA | NQ | PopQA | TriviaQA | HotpotQA | NQ |
| P(True) | 60.00 | 49.65 | 43.34 | 52.83 | 54.58 | 51.76 | 47.73 | 53.78 |
| Logits-mean | 74.89 | 60.24 | 60.18 | 49.92 | 73.47 | 63.46 | 60.35 | 54.89 |
| Logits-max | 58.56 | 52.37 | 52.29 | 46.19 | 56.03 | 54.33 | 48.65 | 48.88 |
| Logits-min | 78.66 | 62.37 | 67.14 | 51.20 | 80.92 | 69.60 | 71.11 | 58.24 |
| Scores-mean | 72.91 | 61.13 | 62.16 | 64.67 | 67.99 | 61.96 | 64.91 | 61.71 |
| Scores-max | 69.33 | 59.74 | 61.29 | 64.08 | 63.34 | 61.92 | 61.09 | 57.56 |
| Scores-min | 64.84 | 55.93 | 59.28 | 55.81 | 61.51 | 56.76 | 63.95 | 57.43 |
| Probing Baseline | 94.25 | 77.17 | 90.25 | 74.83 | 90.96 | 76.61 | 86.54 | 74.20 |
| MoP-RandomGate | 83.69 | 69.20 | 84.11 | 68.76 | 79.69 | 72.38 | 75.13 | 67.11 |
| MoP-VanillaExperts | 93.86 | 78.63 | 90.91 | 75.73 | 90.98 | 77.68 | 86.41 | 75.30 |
| MoP | 95.85 | 80.07 | 91.51 | 79.19 | 92.74 | 78.72 | 88.16 | 78.14 |
| PR | **96.18** | **84.22** | **92.80** | **86.45** | **95.70** | **80.66** | **90.66** | **81.91** |

Table 8: Comparison of hallucination detection performance (AUC) on LLama-3-8B and LLama-3-70B.

| Method | LLama-3-8B | | | | LLama-3-70B | | | |
|---|---|---|---|---|---|---|---|---|
| | PopQA | TriviaQA | HotpotQA | NQ | PopQA | TriviaQA | HotpotQA | NQ |
| P(True) | 55.85 | 49.92 | 52.14 | 53.27 | 54.83 | 50.96 | 49.39 | 51.18 |
| Logits-mean | 74.52 | 60.39 | 51.94 | 52.63 | 67.81 | 52.40 | 50.45 | 48.28 |
| Logits-max | 58.08 | 52.20 | 46.40 | 47.89 | 56.21 | 48.16 | 43.42 | 45.33 |
| Logits-min | 85.36 | 70.89 | 61.28 | 56.50 | 79.96 | 61.53 | 62.63 | 52.16 |
| Scores-mean | 62.87 | 62.09 | 62.06 | 60.32 | 56.81 | 60.70 | 60.91 | 58.05 |
| Scores-max | 56.62 | 60.24 | 59.85 | 56.06 | 55.15 | 59.60 | 57.32 | 51.93 |
| Scores-min | 60.99 | 58.27 | 60.33 | 57.68 | 58.77 | 58.22 | 64.06 | 58.05 |
| Probing Baseline | 88.71 | 77.58 | 82.23 | 70.20 | 86.88 | 81.59 | 84.45 | 74.39 |
| MoP-RandomGate | 75.52 | 69.17 | 79.88 | 66.56 | 67.96 | 70.56 | 72.16 | 66.28 |
| MoP-VanillaExperts | 89.11 | 78.73 | 84.57 | 71.21 | 86.04 | 82.47 | 82.48 | 73.85 |
| MoP | 92.11 | 81.18 | 85.45 | 74.64 | 88.54 | 84.12 | 86.65 | 76.12 |
| PR | **94.01** | **83.13** | **87.81** | **79.10** | **90.08** | **84.21** | **87.69** | **78.24** |

Table 9: Comparison of hallucination detection performance (AUC) on Mistral-7B-v0.1 and Mistral-7B-v0.3.

| Method | Mistral-7B-v0.1 | | | | Mistral-7B-v0.3 | | | |
|---|---|---|---|---|---|---|---|---|
| | PopQA | TriviaQA | HotpotQA | NQ | PopQA | TriviaQA | HotpotQA | NQ |
| P(True) | 48.78 | 50.43 | 51.94 | 55.52 | 45.49 | 47.61 | 57.87 | 52.79 |
| Logits-mean | 69.09 | 64.95 | 54.47 | 59.41 | 69.52 | 66.76 | 55.45 | 57.88 |
| Logits-max | 54.37 | 54.76 | 46.74 | 56.45 | 54.34 | 55.24 | 48.39 | 54.37 |
| Logits-min | 86.02 | 76.56 | 68.06 | 53.73 | 87.05 | 77.33 | 68.08 | 54.40 |
| Scores-mean | 59.00 | 59.61 | 64.18 | 57.60 | 58.84 | 60.22 | 63.28 | 60.05 |
| Scores-max | 51.71 | 56.58 | 63.29 | 55.82 | 53.00 | 55.55 | 63.13 | 57.73 |
| Scores-min | 60.00 | 57.48 | 61.17 | 48.51 | 60.59 | 57.84 | 59.85 | 50.76 |
| Probing Baseline | 89.61 | 78.43 | 83.76 | 74.10 | 87.39 | 81.74 | 83.19 | 73.60 |
| MoP-RandomGate | 80.50 | 68.27 | 74.51 | 68.05 | 79.81 | 70.88 | 72.23 | 61.19 |
| MoP-VanillaExperts | 89.82 | 79.51 | 83.54 | 74.78 | 88.53 | 80.93 | 82.93 | 73.77 |
| MoP | 92.44 | 84.03 | 84.63 | 76.38 | 91.66 | 83.57 | 85.82 | 76.87 |
| PR | **94.72** | **84.66** | **89.04** | **80.92** | **93.09** | **84.36** | **89.03** | **79.09** |

## C.7 RANDOM-TOKEN PATCHING AS A CONTROL CONDITION

To ensure that the effects observed in Section 3.2.3 are not driven by generic perturbations to the input sequence, we conduct a control experiment in which we patch random non-exact tokens instead of exact question tokens. As shown in Figure 15, replacing random tokens produces only minimal

changes in the probe's predictions across both pathways. In contrast, injecting hallucinatory cues into exact question tokens induces a substantially higher flip rate for Q-Anchored samples. This sharp divergence between exact-token and random-token patching confirms that the observed sensitivity arises from the semantic role of exact question tokens in the Q-Anchored pathway rather than from disruption caused by token replacement itself. Consequently, this control analysis strengthens our causal interpretation that exact question tokens are the primary drivers of Q-Anchored pathway activation.

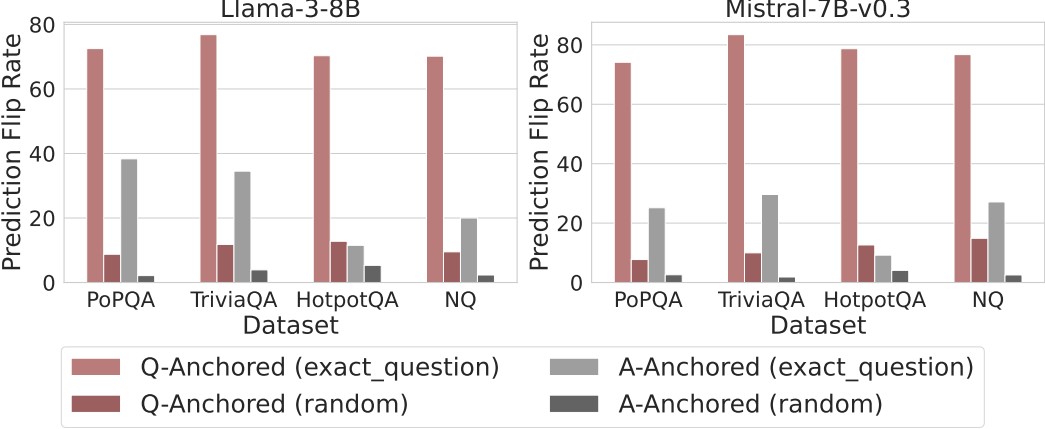

Figure 15: Prediction flip rate under token patching. Q-Anchored samples demonstrate significantly higher sensitivity than the counterparts when hallucinatory cues are injected into exact questions, whereas patching random tokens yields minimal effect.

### C.8  HEAD-WISE QUESTION–ANSWER INFORMATION INTENSITY ANALYSIS

To further examine whether the distinction between Q-Anchored and A-Anchored pathways manifests at a finer granularity, we analyze the question–answer information intensity at the level of individual attention heads. Figures 16 and 17 report the per-head distributions for Llama-3-8B and Mistral-7B-v0.3 across the four datasets used in our analysis.

Within each pathway, different heads exhibit noticeable variability, reflecting their heterogeneous functional roles, which is consistent with observations in prior interpretability work. However, when comparing the two pathways head-by-head, the distributions remain largely homogeneous. In other words, although certain heads may specialize in capturing specific aspects of the question–answer interaction, they do not display pathway-specific shifts in information intensity. These findings suggest that the truthfulness-encoding pathway is not fully aligned with the model's native attention patterns, highlighting a potential space for improving hallucination detection through targeted adjustments to attention mechanisms.

### C.9  ABLATIONS FOR TOKEN SELECTIONS

To assess the robustness of our findings with respect to token selection, we conduct ablations using alternative probing positions. Beyond the exact answer tokens used in the main results, we evaluate two alternatives: (i) the final token of the answer, which is the most common choice in previous approaches given its global receptive field under attention, and (ii) the token immediately preceding the first exact answer token. These settings allow us to test whether the two pathways we identify are sensitive to the specific token chosen for probing.

Figures 18 and 19 report the resulting changes in prediction probability under attention knockout across all layers. Across all three models, the pathway separation persists under both alternative token selections. The layerwise profiles and confidence intervals remain consistent with the patterns observed using exact answer tokens, indicating that the Q-Anchored and A-Anchored pathways are stable under variations in token selections.

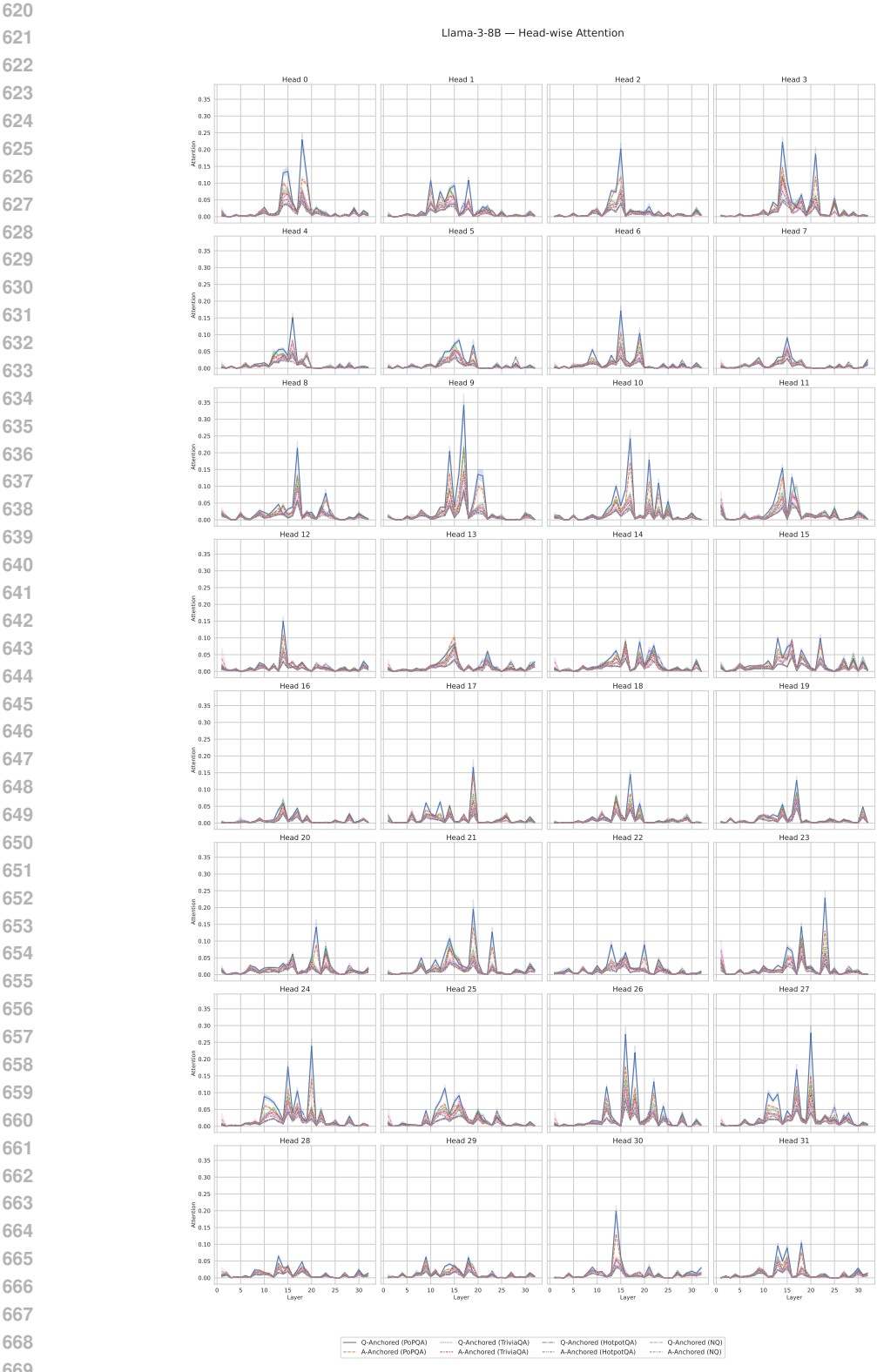

Figure 16: Head-wise question–answer information intensity (Llama-3-8B) for Q-Anchored and A-Anchored samples. Individual heads show noticeable variation, yet each head exhibits highly similar distributions across the two pathways, indicating no pathway-specific differentiation at the head level.

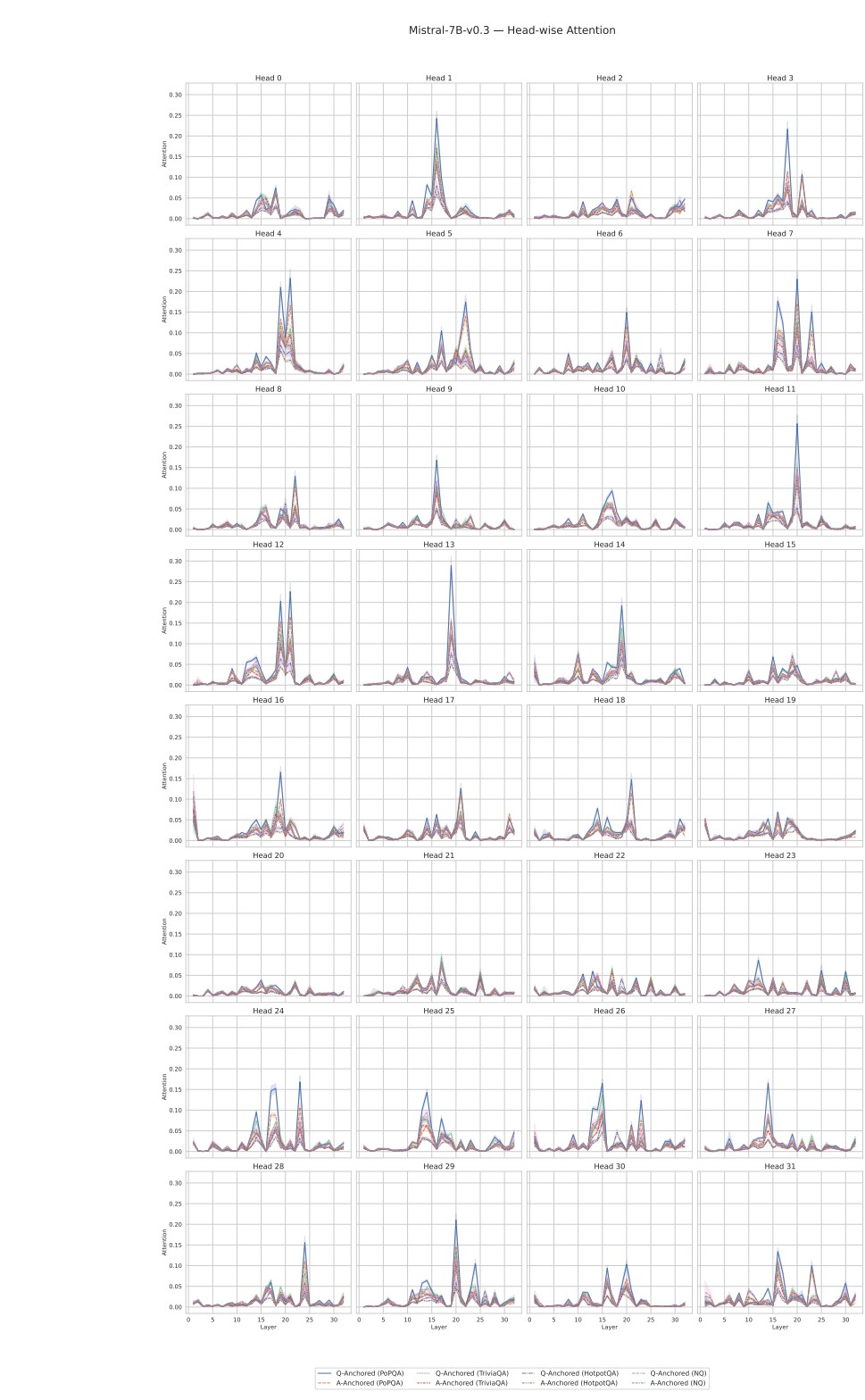

Figure 17: Head-wise question–answer information intensity (Mistral-7B-v0.3) for Q-Anchored and A-Anchored samples. Individual heads show noticeable variation, yet each head exhibits highly similar distributions across the two pathways, indicating no pathway-specific differentiation at the head level.

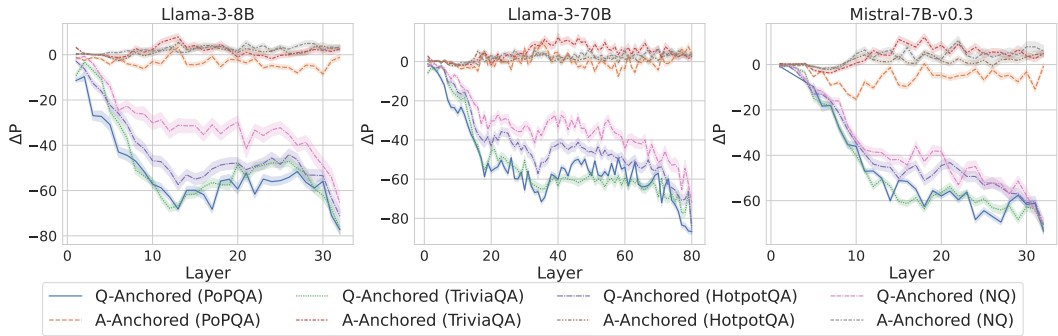

Figure 18: $\Delta$P under attention knockout using the last token in the answer for probing. The layer axis indicates the Transformer layer on which the probe is trained. Shaded regions indicate 95% confidence intervals.

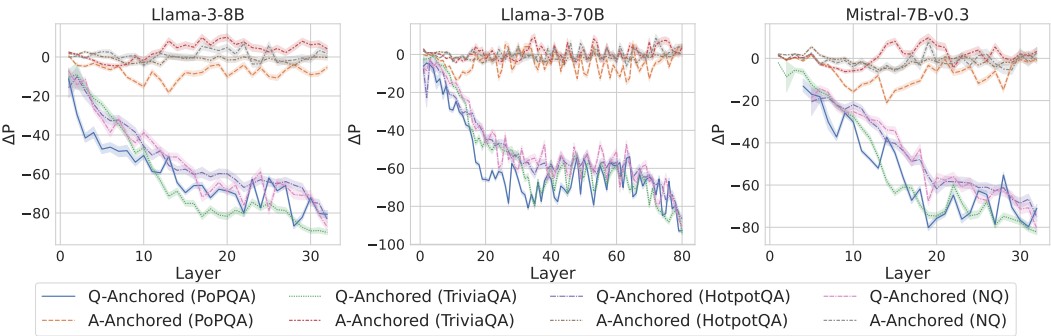

Figure 19: $\Delta$P under attention knockout using the token right before the first exact answer tokens for probing. The layer axis indicates the Transformer layer on which the probe is trained. Shaded regions indicate 95% confidence intervals.

### C.10 LAYER-WISE EFFECTS OF ATTENTION KNOCKOUT

To further examine how different layers contribute to pathway behavior, we conduct a layer-wise attention-knockout experiment using a sliding window of five consecutive layers following common practice (Fierro et al., 2025). For each window, we block the question–answer information flow within those layers and measure the resulting change in pathway predictions ($\Delta$P).

Figure 20 reports the results using the best-performing layer probes. The Answer-Anchored pathway is largely insensitive to knockouts at any layer, consistent with our finding that it does not rely on question–answer interactions. In contrast, the Q-Anchored pathway displays pronounced sensitivity to knockouts in both early and late layers, with weaker effects in the middle layers. This pattern indicates that question–answer information flow in the early and late stages of the forward pass plays a particularly important role in supporting the Q-Anchored pathway.

### C.11 ROBUSTNESS OF PATHWAY STRUCTURE UNDER FINETUNING

To examine whether the two pathways persist after model adaptation, we conduct a finetuning experiment using Llama-3-8B, trained on a isolated subset of the PopQA training set. We then repeat the pathway analysis in Section 3.2.2 by applying attention knockout and measuring the resulting change ($\Delta$P).

As shown in Figure 21, the overall pattern remains highly consistent with the original model. The Q-Anchored pathway continues to show strong sensitivity to knockouts, whereas the A-Anchored pathway remains largely unaffected across all layers. This stability indicates that the underlying computational roles of the two pathways are preserved even after finetuning for factual QA. These

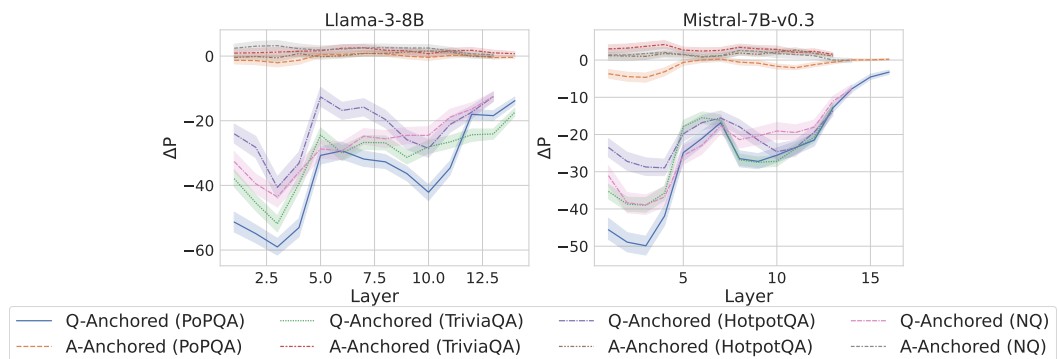

Figure 20: Layer-wise $\Delta P$ under attention knockout. The layer axis indicates the Transformer layer on which attention knockout is conducted. Results using the best-performing layer probes are reported. Shaded regions indicate 95% confidence intervals.

results further support our claim that the pathways represent robust mechanisms rather than artifacts tied to a specific training configuration.

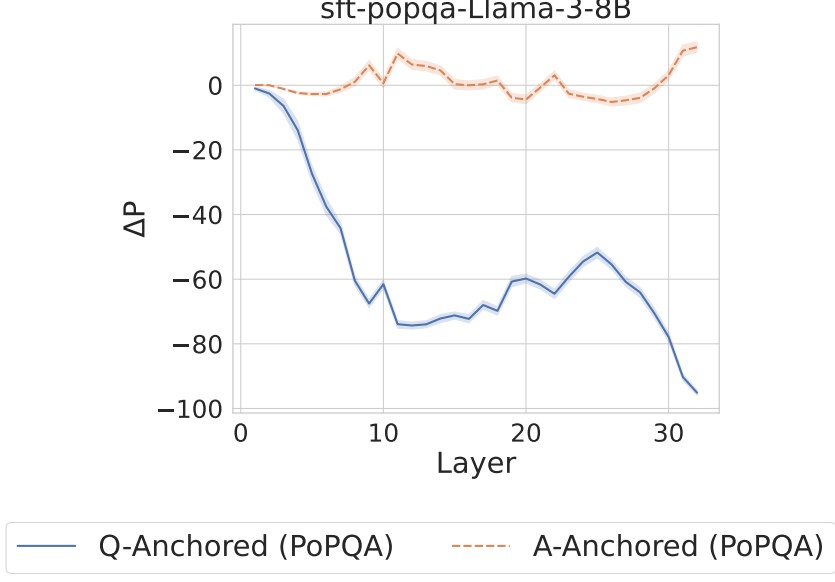

Figure 21: $\Delta P$ under attention knockout after finetuning Llama-3-8B on factual QA. The Q-Anchored pathway retains its characteristic sensitivity to attention knockout, while the A-Anchored pathway remains stable across all layers, demonstrating the robustness of both pathways under factual QA finetuning.

## C.12 ROBUSTNESS OF PATHWAY STRUCTURE UNDER EXTERNAL KNOWLEDGE

To evaluate whether the two pathways persist in the presence of external knowledge, we conduct an experiment on Llama-3-8B using PopQA, augmenting each query with the dataset's built-in retrieved documents, which are the evidence passages PopQA provides for each question. We then repeat the pathway analysis from Section 3.2.2 by applying attention knockout and measuring the resulting change ($\Delta P$).

As shown in Figure 22, the pathway structure remains highly consistent with the setting without external knowledge, indicating that both pathways are robust to the integration of retrieved context.

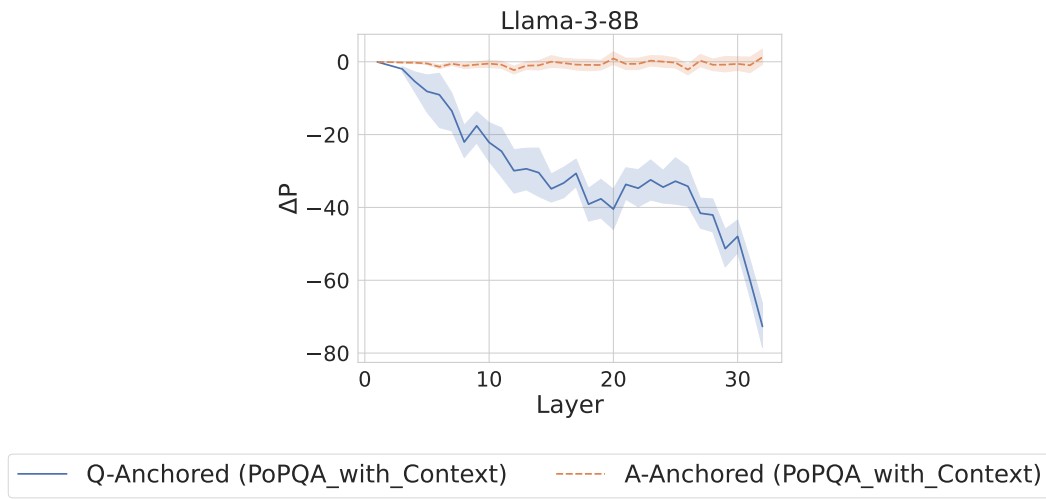

Figure 22: $\Delta P$ under attention knockout when external knowledge is provided as retrieved context. The Q-Anchored pathway retains its characteristic sensitivity to question–answer attention flow, while the A-Anchored pathway remains stable across all layers, demonstrating that the two pathways are robust even when external knowledge is integrated.

### C.13 TOKEN PATCHING BEHAVIOR UNDER NATIVE HALLUCINATIONS

To assess how token patching behaves when applied to native hallucinations rather than to non-hallucinatory ones, we conduct a controlled comparison on Llama-3-8B and Mistral-7B-v0.3. Specifically, we apply token patching separately to (i) correctly answered samples and (ii) natively hallucinated samples across four datasets, using the best-performing probe layers.

As shown in Figure 23, token patching on non-hallucinatory samples consistently yields substantially higher prediction flip rates for both the Q-Anchored and A-Anchored pathways. In contrast, natively hallucinated samples exhibit markedly lower sensitivity. This pattern is expected: once a sample is already in a natively hallucinated state, injecting additional hallucinatory tokens produces only a limited marginal effect, resulting in minimal flip rates.

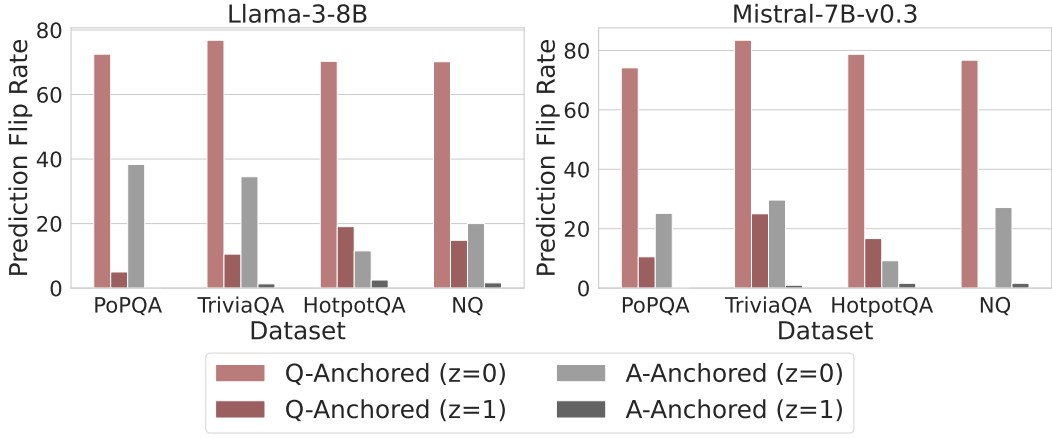

Figure 23: Prediction flip rates under token patching. Across all datasets and both pathways, non-hallucinatory samples exhibit substantially higher sensitivity to token patching compared to natively hallucinated samples.

## C.14 FINER-GRAINED METRICS FOR KNOWLEDGE BOUNDARIES

To further refine our characterization of knowledge boundaries, we incorporate a finer-grained metric based on entity popularity. Following prior work (Mallen et al., 2023), we use standardized popularity scores that are widely adopted to distinguish well-known entities from long-tail factual knowledge.

We evaluate this metric across four models on PopQA and compare the popularity distributions between Q-Anchored and A-Anchored samples. As shown in Figure 24, Q-Anchored samples tend to involve more popular entities, whereas A-Anchored samples are associated with less popular, long-tail entities. This trend is consistent with our observations in Section 4.1: Q-Anchored cases typically lie within the model's knowledge boundary, while A-Anchored cases exhibit higher uncertainty and a greater tendency toward hallucination.

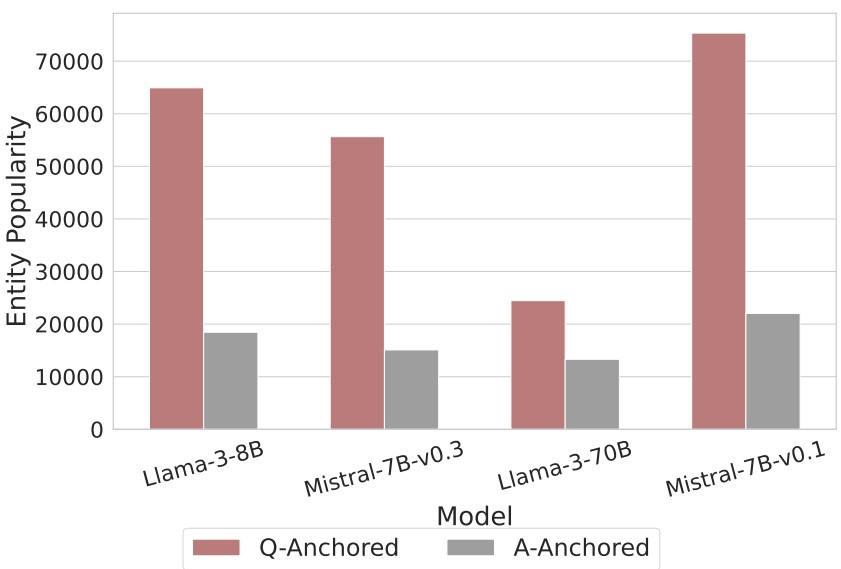

Figure 24: Entity popularity distributions for Q-Anchored and A-Anchored samples across four models on PopQA. Q-Anchored samples concentrate on more popular entities, whereas A-Anchored samples skew toward long-tail entities

