# OpenReview forum: "Two Pathways to Truthfulness: On the Intrinsic Encoding of LLM Hallucinations"
_ICLR.cc/2026/Conference — ICLR 2026 Conference Withdrawn Submission_

### Official Review · Reviewer_YZZQ · 2025-10-29

**Soundness:** 3
**Presentation:** 3
**Contribution:** 2
**Rating:** 4
**Confidence:** 3

**Summary:**

This paper investigates how large language models internally encode truthfulness signals to detect their own hallucinations in question-answering tasks. Through saliency analysis on attention flows, the authors identify a bimodal distribution suggesting two separate pathways: a Question-Anchored pathway that relies on information flowing from exact question tokens to the answer, and an Answer-Anchored pathway that uses self-contained cues within the generated answer. They validate this distinction using attention knockout to block question-to-answer flows and token patching to inject hallucinatory question cues, showing sharp behavioral splits across Llama-3 and Mistral models on PopQA, TriviaQA, HotpotQA, and Natural Questions datasets. Further analyses link Question-Anchored encoding to high-confidence known facts and Answer-Anchored to uncertain or extrapolated responses, demonstrate that models can predict which pathway is active from internal states, and reveal that pathway selection misaligns with uniform question-answer attention during standard generation. The insights enable two new detection methods—Mixture-of-Probes that routes to pathway-specific classifiers and Pathway Reweighting that amplifies salient signals—yielding up to 10% AUC gains.

**Strengths:**

The core contribution of disentangling two truthfulness pathways is novel and rigorously substantiated; attention knockout produces clean, statistically robust bifurcations in prediction shifts, while token patching elegantly confirms differential sensitivity to question-derived hallucinations. The large-scale scope across six model sizes from two families and four diverse datasets strengthens generalizations beyond prior probing studies that treat truthfulness signals as monolithic.

Linking pathways to knowledge boundaries via answer accuracy and generation entropy provides interpretable ties to model capabilities, and the self-awareness probe achieving high accuracy in predicting pathway usage reveals sophisticated internal monitoring. The proposed Mixture-of-Probes and Pathway Reweighting directly exploit these mechanisms for tangible gains in hallucination detection, outperforming layer-wise probing baselines consistently. Focus on exact question and answer tokens grounded in semantic frame theory sharpens analysis compared to whole-sequence averaging.

**Weaknesses:**

Attention knockout is applied cumulatively up to layer k without isolating per-layer contributions or controlling for indirect flows through intermediate tokens, potentially conflating direct and propagated effects. Token patching experiments restrict to correct-answer contexts with non-hallucination predictions, limiting insight into how pathways behave under native hallucinations rather than injected ones. Knowledge boundary associations rely on answer accuracy and entropy but ignore finer-grained metrics like memorization versus reasoning or entity frequency in pretraining, leaving the "boundary" definition coarse.

Self-awareness results train the pathway predictor on the same probing layers used for hallucination detection, risking circularity since the predictor may simply rediscover the knockout intervention signal.

The applications evaluate only on the same four datasets used for discovery, with no held-out tasks like long-form generation or multi-hop reasoning where pathway dynamics might differ. Ablations for Mixture-of-Probes do not compare against simpler ensembles, and Pathway Reweighting lacks clarity on how reweighting coefficients are derived beyond "information intensity."

**Questions:**

How sensitive are the bimodal saliency distributions to the choice of exact token identification—does switching to dependency-parse heads or named entities alter the peaks?

In attention knockout, why does prediction stability for Answer-Anchored instances hold even in early layers where question context is still accessible, and what fraction of information reroutes through non-exact question tokens?

For token patching, what happens when patching hallucinatory cues into Answer-Anchored instances that are already hallucinations—does the probe flip less or more than in correct cases?

---

> ### Author Response · Authors · 2025-11-21
> **Response to Reviewer YZZQ Part 1/2**
>
> We thank the reviewer for the constructive suggestions and attentive review. Below, we provide our responses to each comment.
>
> > **per-layer contributions and controlling for indirect flows**
>
> 1. In response to the reviewer’s suggestion, we performed a layer-wise attention-knockout analysis to more precisely assess how per-layer question–answer information flow contributes to each pathway. Following common practice [1], we use a sliding window of five consecutive layers and apply attention knockout within each window. **Appendix C.10 presents the results using the best-performing layer probes. For the Answer-Anchored pathway, knocking out any layer yields no noticeable effect, consistent with our earlier finding that this pathway does not depend on question–answer flow. In contrast, for the Q-Anchored pathway, knockouts in both early and late layers produce substantial impact, whereas middle-layer knockouts exhibit weaker effects.**
>
> 2. For controlling indirect flows through intermediate tokens, **it is important to clarify that our attention-knockout method explicitly masks all attention edges from exact question tokens to any subsequent tokens, thereby eliminating both direct and indirect information transfer along question–answer attention paths.**
>
> > **token patching under native hallucinations**
>
> Additional experiments were conducted to compare token patching on non-hallucinated versus natively hallucinated examples. We applied token patching separately to correctly answered samples and to hallucinated samples across four datasets using Llama-3-8B and Mistral-0.3. **As shown in Appendix C.13, for both models and both pathways, token patching on non-hallucinated examples yields substantially higher flip rates than on hallucinated ones. This outcome is expected because natively hallucinated examples are already erroneous, so introducing further hallucinatory perturbations leads to a minimal effect.**
>
> > **finer-grained metrics for knowledge boundaries**
>
> We appreciate the reviewer’s suggestion to incorporate finer-grained metrics into our knowledge-boundary analysis. To address this, we conducted an additional study using entity popularity as a more granular indicator, following prior work [2] where popularity scores are widely used to distinguish between common and long-tail factual knowledge.
>
> **As shown in Appendix C.14, across four models on PopQA, we find that Q-Anchored samples generally involve more popular entities, whereas A-Anchored samples are associated with less popular, long-tail entities.** This pattern is consistent with our earlier observations in Section 4.1: Q-Anchored samples tend to lie near the model’s knowledge boundary, while A-Anchored samples reflect higher uncertainty.
>
> > **clarification on self-awareness probes**
>
> We would like to clarify that for training the self-awareness probes, **we re-run a separate forward pass without any attention knockout to obtain the vanilla hidden states.** This ensures that the probes are trained on the original model representations, free from any artifacts introduced by the knockout intervention.
>
> > **evaluation on long-form or multi-hop tasks**
>
> We fully agree with the reviewer on the importance of assessing our framework beyond standard QA formats. To address this, we evaluate our applications in a long-form, open-ended reasoning scenario using Qwen3-8B on HotpotQA, which requires multi-hop reasoning. **MoP achieves an AUC of 83.08% and PR attains 84.17%, both substantially outperforming the probing baseline’s AUC of 79.26%.** These results demonstrate that our framework remains effective in long-form, open-ended generation scenarios.
>
> > **supplementary ablations and clarification of PR coefficients**
>
> 1. For MoP ablations, we have added two variants: (1) MoP-RandomGate, which randomly routes the two pathway experts without using the self-awareness probe, and (2) MoP-VanillaExperts, which uses two vanilla probes to replace the specialized Q-Anchored and A-Anchored probes. **As shown in Appendix C.6, both ablated models perform significantly worse than MoP, confirming the importance of both pathway specialization and self-awareness gating.**
>
> 2. For PR, in the original paper, we have already clarified that the reweighting coefficients are two learnable parameters that scale the question-answer attention flow for Q-Anchored and A-Anchored samples. These learnable coefficients are optimized alongside the probe during training to maximize hallucination detection performance.

---

> ### Author Response · Authors · 2025-11-21
> **Response to Reviewer YZZQ Part 2/2**
>
> > **sensitivity to exact token identification choices**
>
> In Appendix C.9, we report results under different token selections. Specifically, beyond the exact answer tokens, we also use **(1) the token immediately preceding the exact answer span** and **(2) the last token of the answer** (a common choice in previous approaches given its global receptive field under attention). **Results in Appendix C.9 demonstrate the robustness of the two pathways regardless of which tokens are used.**
>
> > **A-Anchored stability and possible rerouted information**
>
> 1. Because our knockout intervention removes all attention from the exact question tokens to every subsequent token across all layers, the model has no access to the question context after the intervention. The resulting stability of the Answer-Anchored pathway is therefore a key empirical finding, showing that this pathway does not depend on any question–answer attention flow.
>
> 2. We would appreciate clarification from the reviewer regarding the intended meaning of “the fraction of information rerouting through non-exact question tokens,” as we want to ensure we fully understand the concern in order to address it appropriately.
>
> > **effect of injecting hallucinatory cues into already-hallucinating A-Anchored cases**
>
> Please refer to **token patching under native hallucinations in Part 1.**
>
>
>
> **References:**
>
> [1] Fierro, Constanza, et al. "How Do Multilingual Language Models Remember Facts?." Findings of the Association for Computational Linguistics: ACL 2025.
>
> [2] Mallen, Alex, et al. "When not to trust language models: Investigating effectiveness of parametric and non-parametric memories." Proceedings of the 61st Annual Meeting of the Association for Computational Linguistics (Volume 1: Long Papers). 2023.

---

> ### Author Response · Authors · 2025-11-28
> **Your feedback is appreciated**
>
> Dear Reviewer,
>
> Thank you again for your time and effort in reviewing our paper. We have clarified the questions raised and hope our rebuttal have addressed your concerns. Please don’t hesitate to let us know if you have any further questions.

---

### Official Review · Reviewer_vwQd · 2025-10-31

**Soundness:** 4
**Presentation:** 4
**Contribution:** 3
**Rating:** 6
**Confidence:** 3

**Summary:**

This paper investigates the intrinsic truthfulness encoding mechanisms of large language models (LLMs), uncovering two distinct information pathways: Question-Anchored (relying on question-answer information flow) and Answer-Anchored (deriving evidence from generated answers themselves). Through attention knockout, token patching, and large-scale experiments across four datasets and multiple model architectures (Llama-3, Mistral-7B), the paper identifies three key properties of these pathways—association with knowledge boundaries, intrinsic self-awareness, and misalignment with language modeling objectives. Building on these findings, two pathway-aware hallucination detection methods (Mixture-of-Probes and Pathway Reweighting) are proposed, achieving up to 10% AUC gain compared to baselines. The work deepens the understanding of LLM hallucinations and provides practical directions for building more reliable generative systems.

**Strengths:**

* **Novel Mechanistic Insight**: The idea of decomposing truthfulness encoding into two interpretable pathways is original and provides a new conceptual framework for understanding hallucinations.
* **Methodological Rigor**: The use of attention knockout, token patching, and multi-model, multi-dataset validation is impressive and methodologically sound.
* **Empirical Significance**: The consistent performance gains of MoP and PR demonstrate strong practical value beyond theoretical contributions.
* **Clarity and Presentation**: The paper is clearly written, with well-structured figures and a logical flow from hypothesis to validation to application.

**Weaknesses:**

* The paper could further **clarify the causal direction** between pathway activation and hallucination occurrence. While associations are well-demonstrated, causal inference remains somewhat implicit.
* The **interpretability of “self-awareness”** could be discussed with more nuance — e.g., whether this phenomenon indicates metacognitive capability or simply separable representational clusters.
* **Comparative analysis** with existing mechanistic interpretability frameworks (e.g., Transformer Circuits, causal tracing) is limited; positioning the work within that literature would strengthen its conceptual grounding.
* The **applications section (Section 5)**, though interesting, feels condensed compared to the theoretical parts; additional ablations on gating accuracy or scalability would improve completeness.

**Questions:**

1. How sensitive are the identified pathways to model size or instruction tuning—would finetuning for factual QA alter the pathway balance?
2. Could the authors elaborate on whether “A-Anchored” signals are related to self-consistency or internal entailment mechanisms seen in reasoning tasks?
3. How does the MoP framework generalize to non-QA settings (e.g., open-ended generation or summarization)?
4. Would integrating external knowledge retrieval alter or reinforce the two-pathway distinction?

---

> ### Author Response · Authors · 2025-11-21
> **Response to Reviewer vwQd**
>
> We are grateful to the reviewer for the insightful remarks and the time invested in evaluating our work. Below, we provide our responses to each comment.
>
> > **causal direction between pathway activation and hallucination occurrence**
>
> We appreciate the reviewer’s suggestion. Our main claim is that the two identified pathways causally encode truthfulness, as supported by the attention-knockout and token-patching experiments in Section 3. Section 4.1 further reports several associations between pathway activation and hallucination occurrence.
>
> We fully agree that establishing a direct causal link from pathway activation to hallucination occurrence is important. However, it is still challenging, as hallucinations may arise from complex interactions among multiple model components and inputs. Nonetheless, we believe our results offer initial evidence toward this relationship, and we are open to expanding this direction in future work. Thanks again for the valuable feedback.
>
> > **nuanced interpretation of self-awareness**
>
> We thank the reviewer for raising this important point. We agree that the interpretability of “self-awareness” could be discussed with more nuance. Given the current scope of our work, we primarily focus on demonstrating the existence of separable representational clusters corresponding to the two pathways. We acknowledge that further investigation is needed to verify whether these clusters reflect metacognitive capabilities. We will clarify this in the revised version and are willing to explore this aspect in future research.
>
> > **comparison with existing mechanistic interpretability frameworks**
>
> We will expand our related work section to better position our contributions within the broader mechanistic interpretability literature, including Transformer circuits and causal tracing. We believe that our focus on identifying intrinsic pathways for truthfulness encoding in LLMs complements these existing frameworks, and we will make further clarifications in the revised manuscript.
>
> > **expand applications with gating performance**
>
> **In the original paper, we directly use the self-awareness probe from Section 4.2 without additional training, so the gating performance is already demonstrated in Section 4.2.**
>
> > **pathway sensitivity to finetuning**
>
> We have conducted additional experiments to assess the sensitivity of the identified pathways to finetuning. Specifically, we finetuned Llama-3-8B on a isolated subset of PopQA training set, and then conducted the same pathway analysis as in Section 3.2.2 via attention knockout. **As shown in Appendix C.11, two identified pathways remain robust after finetuning for factual QA.** These results further support our claim that the pathways represent robust mechanisms rather than artifacts tied to a specific training configuration.
>
> > **relation between pathway signals and self-consistency mechanisms**
>
> We have analyzed the consistency in generations under the two pathways using PopQA with Llama-3-8B. Specifically, for each question, we sample ten answers using temperature sampling (temperature=0.6). Then, for both Q-Anchored and A-Anchored conditions, exact answers are identified, and the majority exact answer frequency is computed as a measure of generation consistency.
>
> **Results show that Q-Anchored samples achieve 50.30% consistency, while A-Anchored samples reach only 35.98% consistency, suggesting that A-Anchored samples tend to produce less consistent generations.** This result is aligned with Section 4.1, where A-Anchored samples exhibit higher uncertainty than Q-Anchored ones.
>
> > **generalization of MoP**
>
> It is notable that in the original paper, we allow unrestricted generation for models to produce open-ended answers, rather than constraining them to extractive QA.
>
> We fully agree on the importance of assessing the generalizability of MoP beyond standard QA formats. To this end, we additionally evaluated MoP in a long-form, open-ended reasoning setting using Qwen3-8B on HotpotQA, which requires multi-hop reasoning. **MoP achieves an AUC of 83.08%, compared to 79.26% for the probing baseline, demonstrating its effectiveness in long-form generation scenarios.**
>
> > **impact of external knowledge on pathway distinction**
>
> To assess whether external knowledge retrieval affects the two pathways, we conducted additional experiments on Llama-3-8B using PopQA, utilizing the dataset’s built-in retrieved documents, which are the evidence passages PopQA provides for each question. We then applied the same attention-knockout pathway analysis as in Section 3.2.2.
>
> **As shown in Appendix C.12, the two identified pathways remain robust when external knowledge is integrated.**

---

> ### Author Response · Authors · 2025-11-28
> **Your feedback is appreciated**
>
> Dear Reviewer,
>
> Thank you again for your time and effort in reviewing our paper. We have clarified the questions raised and hope our rebuttal have addressed your concerns. Please don’t hesitate to let us know if you have any further questions.

---

### Official Review · Reviewer_5BX9 · 2025-11-01

**Soundness:** 3
**Presentation:** 3
**Contribution:** 2
**Rating:** 4
**Confidence:** 4

**Summary:**

This paper examines how LLMs internally encodes truthfulness, aiming to uncover the mechanisms underlying intrinsic hallucination-detection signals in their representations. The authors propose that two distinct internal “truthfulness pathways” underlie these signals: the Question-Anchored Pathway and the Answer-Anchored Pathway. To identify and disentangle two pathways, the paper blocks question–answer attention and token patching. The experiments, conducted on multiple datasets (PopQA, TriviaQA, HotpotQA, NQ) and models (Llama-3 1B, 3B, 8B, 70B, Mistral-7B), reveal a bimodal distribution of dependency on question–answer interactions—supporting the existence of two separate truthfulness mechanisms.

The author further applied their experiment findings to hallucination detection in two ways:  1) Mixture-of-Probes (MoP): Specialized classifiers for each pathway combined via a gating network predicting pathway usage. 2) Pathway Reweighting (PR): Adjusts attention flows to amplify pathway-relevant signals for hallucination detection.

**Strengths:**

1. The motivation and findings are interesting. First, the author found that Q-Anchored signals align with knowledge boundaries and they dominate when the model is confident and knowledgeable. Second, LLMs exhibit intrinsic self-awareness that internal representations can predict which pathway is active.

2. The author uses multiple interpretability methods (attention knockout, token patching, answer-only experiments) to triangulate evidence.  The author also tested in various model families and parameter size (Llama-3 and Mistral, from 1B to 70B), four QA datasets, and multiple scales, demonstrating the generalizability of the findings.

3. The paper is well-written and easy to follow.

**Weaknesses:**

1. The entire experiment analysis relies on identifying "exact question" and "exact answer" tokens using GPT-4o. This step relies on a proprietary model and lacks a detailed explanation of the experiment setup here. That may change over time. Also, no further ablation study on sensitivity to token selection choices.

2. Insufficient explanation of the pathway selection mechanism. Although the paper shows the model can predict which pathway is used, I still have several questions about this step. Is selection deterministic or probabilistic? And how early in the forward pass is pathway selection determined?

3. For the patching, how does the author ensure the patch sample is comparable in difficulty? Random selection could introduce noise if patch questions are easier or harder.

**Questions:**

See in Weaknesses.

---

> ### Author Response · Authors · 2025-11-21
> **Response to Reviewer 5BX9**
>
> We appreciate the reviewer’s careful reading and valuable feedback. Below, we provide our responses to each comment.
>
> > **token-identification clarity and selection ablation**
>
> 1. For token identification, prior work [1] has shown that LLMs can accurately identify exact answer tokens, typically achieving over 95% accuracy. In addition, we manually verified GPT-4o’s identification quality in our setting. Specifically, it achieves 99.92%, 95.83%, and 96.62% accuracy on exact subject tokens, exact property tokens, and exact answer tokens, respectively. We will include these results in the revised version for clarity.
>
> 2. For the ablation on token selection, we added experiments where probing is performed on alternative token positions. Beyond the exact answer tokens, we also use **(1) the token immediately preceding the exact answer span** and **(2) the last token of the answer** (a common choice in previous approaches given its global receptive field under attention). **Appendix C.9 reports results under these different token selections, showing that the two pathways remain stable regardless of which tokens are used.**
>
> > **further clarification on pathway mechanisms**
>
> We have added a layer-wise attention-knockout analysis to further examine how per-layer question–answer information flow contributes to each pathway. Following common practice [2], we apply a sliding window of five consecutive layers and perform attention knockout within each window. **Appendix C.10 reports the results using the best-performing layer probes. For the Answer-Anchored pathway, knocking out any layer yields no noticeable effect, consistent with our earlier finding that this pathway does not rely on question–answer flow. In contrast, for the Q-Anchored pathway, knockouts in both early and late layers have substantial impact, whereas middle-layer knockouts show weaker effects.**
>
> **These results suggest a largely deterministic pathway selection: for some samples, the model encodes truthfulness purely from its self-generated output representations, while for others, it relies on question–answer interactions.** Moreover, the early and late layers’ question–answer interactions appear particularly important for the Q-Anchored pathway. This observation can be better understood in light of our findings in Section 4.1, which provide additional context showing that Q-Anchored samples often lie near the model’s knowledge boundary, whereas A-Anchored samples exhibit higher uncertainty.
>
> We will incorporate these results into the revised version and plan to investigate the mechanism in greater depth in future work.
>
> > **control for patch-sample difficulty**
>
> As a control, we include a random-token patching experiment. Following Section 3.2.3, we randomly select patch examples and replace non-exact question tokens. **As shown in Appendix C.7, random-token patching has a much smaller effect on both the Q-Anchored and A-Anchored pathways compared to exact-question-token patching.** This rules out general noise from token replacement and patch-example selection.
>
> **References:**
>
> [1] Orgad, Hadas, et al. "LLMs Know More Than They Show: On the Intrinsic Representation of LLM Hallucinations." The Thirteenth International Conference on Learning Representations.
>
> [2] Fierro, Constanza, et al. "How Do Multilingual Language Models Remember Facts?." Findings of the Association for Computational Linguistics: ACL 2025.

---

> ### Author Response · Authors · 2025-11-28
> **Your feedback is appreciated**
>
> Dear Reviewer,
>
> Thank you again for your time and effort in reviewing our paper. We have clarified the questions raised and hope our rebuttal have addressed your concerns. Please don’t hesitate to let us know if you have any further questions.

---

### Official Review · Reviewer_4YZJ · 2025-11-01

**Soundness:** 2
**Presentation:** 3
**Contribution:** 3
**Rating:** 4
**Confidence:** 4

**Summary:**

This paper investigates the mechanisms underlying truthfulness encoding in large language models (LLMs) during hallucination detection. The authors discover two distinct information pathways: (1) a Question-Anchored (Q-Anchored) pathway that relies on question-answer information flow, and (2) an Answer-Anchored (A-Anchored) pathway that derives evidence from the generated answer itself. Through attention knockout and token patching experiments across multiple models, the authors present evidence for these pathways. Building on these findings, the authors further propose two hallucination detection methods that leverage the pathway information.

**Strengths:**

- The presentation of the paper is good. The authors present their findings in a clear manner and include necessary details and supporting evidence.

- The paper illustrates the practical value of the findings through two hallucination detection algorithms guided by the pathway insights.

- The structure of the paper is good. The authors present a complete research process: initial observation, hypothesis formulation, experimental investigation, and derivation of practical applications.

**Weaknesses:**

- Suggestion for additional analysis: In Section 3.1, the authors present saliency score distributions aggregated across all samples, revealing a bimodal pattern. As a manipulation check, it would be valuable to stratify this analysis by the actual presence of hallucination (z=1 vs. z=0). Specifically, showing saliency patterns separately for hallucinated versus truthful responses could reveal whether different pathways are preferentially engaged depending on whether hallucination is actually occurring. This could provide deeper insights into the relationship between the detection mechanisms and the phenomenon being detected, and potentially uncover additional patterns about when and why each pathway is utilized.

- The token patching experiment (Section 3.2.3) would benefit from a control condition where random (non-exact) tokens are replaced instead of exact question tokens. This would rule out the alternative explanation that prediction changes result from general disruption due to token replacement rather than specifically from the semantic content of exact question tokens. Such a control would strengthen the causal claim that exact question tokens drive the Q-Anchored pathway.

- The authors claim "misalignment between truthfulness encoding and language modeling" based on homogeneous aggregated attention patterns across Q-Anchored and A-Anchored pathways (Figure 6). This conclusion is premature for several reasons:
  - Aggregation masks important differences: Averaging attention across all heads may obscure head-specific patterns. Prior work demonstrates that different attention heads serve distinct functions.  Head-level analysis is needed to determine whether specific heads show differential patterns between pathways.
  - Attention may not be the relevant mechanism: The pathway distinction might manifest in other components rather than in attention weights. The authors should analyze whether these other mechanisms show pathway-dependent differences.
  - Mechanism vs. artifact: Fundamentally, it's unclear whether these pathways represent actual computational mechanisms used during generation, or merely patterns visible through the probe's lens. That attention patterns (which drive generation) don't distinguish pathways, while probe predictions (which don't affect generation) do raise the question: are these pathways genuinely mechanistic, or post-hoc observational artifacts?

**Questions:**

- In Section 3.2.4, please clarify whether the hidden states analyzed are re-obtained by running a new forward pass with only the answer as input, or whether the original hidden states (from processing the full question+answer) are used. This distinction is crucial for interpreting the 'answer-only' condition, as original hidden states would contain implicit question information through the attention mechanism.

---

> ### Author Response · Authors · 2025-11-21
> **Response to Reviewer 4YZJ**
>
> We sincerely thank the reviewer for the thoughtful comments and careful evaluation. Below, we provide our responses to each comment.
>
> > **stratify patterns by hallucination vs. non-hallucination**
>
> In our saliency analysis, our primary goal was to motivate the presence of two distinct information pathways that encode truthfulness, so we did not further separate hallucinated and truthful samples. We agree that such stratification could offer additional insights. **As discussed in Section 4.1 in the original paper, we already analyze hallucination occurrence under the two pathways in details**, and the results show that Q-Anchored samples exhibit significantly lower hallucination rates (i.e., higher accuracy) compared to those driven by the A-Anchored pathway.
>
> > **non-exact token patching**
>
> We have added the random-token patching control. **As shown in Appendix C.7, patching random tokens produces much weaker effects than patching exact question tokens, for both the Q-Anchored and A-Anchored pathways.** This rules out general disruption from token replacement and further supports our causal claim that exact question tokens drive the Q-Anchored pathway.
>
> > **head-wise attention analysis**
>
> 1. In response to the suggestion for more fine-grained analysis, we have conducted head-level examinations for both Llama3-8B and Mistral-0.3 across four datasets. **As shown in Appendix C.8, individual heads within each pathway do exhibit distinct functional patterns. However, when comparing Q-Anchored and A-Anchored pathways head-by-head, their distributions remain highly similar, indicating that although specific heads may specialize in certain roles, they do not show differential patterns between the two pathways.**
>
> 2. In Section 3.2.2, we use attention knockout to demonstrate the causal role of the question–answer attention flow in driving the two pathways. Furthermore, Section 5.2 shows that modulating this flow can substantially improve hallucination detection performance, reinforcing the importance of the attention mechanism in both pathways. By referring to “misalignment,” we mean that there is potential to adjust the attention mechanism to yield more favorable representations for both pathways and thereby enhance hallucination detection, as demonstrated in Section 5.2. We will further clarify these points in the revised version.
>
> > **Clarify how hidden states for the answer-only condition are obtained**
>
> **In the answer-only condition, we re-run a separate forward pass using only the answer as input.** This ensures that the resulting hidden states do not contain implicit question information introduced by attention in the original question–answer forward pass.

---

> ### Author Response · Authors · 2025-11-28
> **Your feedback is appreciated**
>
> Dear Reviewer,
>
> Thank you again for your time and effort in reviewing our paper. We have clarified the questions raised and hope our rebuttal have addressed your concerns. Please don’t hesitate to let us know if you have any further questions.

---

### Author Response · Authors · 2025-11-21
**General Response to Reviewers**

We sincerely thank all reviewers for their careful reading of our manuscript and for the constructive feedback. We have provided detailed, point-by-point responses to every question and suggestion raised. **Because many of the newly added analyses and experiments are best communicated visually, the corresponding detailed results are presented in figures in the revised appendix. All modifications are clearly marked in red for ease of reference.**

---

### Author Response · Authors · 2025-11-28
**Kind Reminder Regarding Our Rebuttal**

Dear Reviewers,

Thank you very much for your thoughtful and constructive comments on our submission. We have carefully addressed each point raised and provided detailed clarifications and updates in our rebuttal.

As the discussion deadline is approaching, we kindly ask whether you could take a moment to review our responses at your convenience. We would greatly appreciate any follow-up feedback regarding whether our clarifications sufficiently resolve your concerns.

Thank you again for your time and effort in reviewing our work. We sincerely value your input.

Best regards,

The Authors

---

### Note · Authors · 2025-12-03

I have read and agree with the venue's withdrawal policy on behalf of myself and my co-authors.